



# Re-evaluating the 1940s CO$_2$ plateau

Ana Bastos[1], Philippe Ciais[1], Jonathan Barichivich[1], Laurent Bopp[1], Victor Brovkin[2], Thomas Gasser[1], Shushi Peng[3], Julia Pongratz[2], Nicolas Viovy[1], and Cathy M. Trudinger[4]

[1]Laboratoire des Sciences du Climat et de l'Environnement, LSCE/IPSL, CEA-CNRS-UVSQ, Université Paris-Saclay, F-91191 Gif-sur-Yvette, France.
[2]Max Planck Institute for Meteorology, Bundesstraße 53, 20146 Hamburg, Germany.
[3]Sino-French Institute for Earth System Science, College of Urban and Environmental Sciences, Peking University, Beijing 100871, China.
[4]CSIRO Oceans and Atmosphere, Aspendale, Victoria, Australia

*Correspondence to:* Ana Bastos (ana.bastos@lsce.ipsl.fr)

**Abstract.** The high-resolution CO$_2$ record from Law Dome ice core reveals that atmospheric CO$_2$ concentration stalled during the 1940s (so-called CO$_2$ plateau). This stalling implies the persistence of a sink of the same magnitude as the concurrent fossil fuel emissions, perhaps sustained for as long as a decade or more. This sink has been previously attributed to the ocean, conceivably as a response to the very strong El Niño event in 1940-42. However, this explanation is questionable, as recent

ocean CO$_2$ data indicate that the range of variability in the ocean sink has been rather modest in recent decades, and El Ni no events have generally led to higher growth-rates of atmospheric CO$_2$ due to the offsetting terrestrial response. Here, we use the most up-to-date information on the different terms of the carbon budget: fossil fuel emissions, four estimates of land-use change (LUC) emissions, ocean uptake from two different reconstructions, and the terrestrial sink modelled by the TRENDY project. Evaluating whether these datasets provide further insight about the 1940s plateau and its causes, we find that, they give

a plausible explanation for most of the 20th century carbon budget, especially from 1970 onwards, but they greatly overestimate atmospheric CO$_2$ growth rate during the plateau period, as well as in the 1960s. The mismatch between reconstructions and observations during the CO$_2$ plateau epoch of 1940-1950 ranges between 0.9-2.0 Pg C yr$^{-1}$, depending on the LUC dataset considered. This mismatch may be explained by: i) decadal variability in the ocean carbon sink not accounted for in the reconstructions we used; ii) a further terrestrial sink currently missing in the estimates by land-surface models; iii) land-use

change processes not included in the current datasets. Ocean carbon models from CMIP5 indicate that natural variability in the ocean carbon sink could explain an additional 0.5 Pg C yr$^{-1}$ uptake, but it is unlikely to be higher. The impact of the 1940-42 El Ni no on the observed stabilization of atmospheric CO$_2$ cannot be confirmed nor discarded, as TRENDY models do not reproduce the expected concurrent strong decrease in terrestrial uptake. Nevertheless, this would further increase the mismatch between observed and modelled CO$_2$ growth rate during the CO$_2$ plateau epoch. Tests performed using the OSCAR (v2.2)

model, indicate that changes in land use not correctly accounted for during the period (coinciding with drastic socioeconomic changes during WW2) could contribute to the additional sink required.Thus, the previously proposed ocean hypothesis for the 1940s plateau cannot be confirmed by independent data. Further efforts are required to reduce uncertainty in the different terms of the carbon budget during the first half of the 20th century, and to better understand the long-term variability of the ocean and terrestrial CO$_2$ sinks.





# 1 Introduction

Study of the long-term variability in atmospheric composition from air trapped in polar ice has improved our understanding of processes and feedbacks between climate and the carbon cycle on decadal to millennial scales, and allows us to evaluate of the magnitude of human impact on the Earth's atmosphere. Since the mid-20th century, atmospheric $CO_2$ monitoring has progressed from the first consistent measurements in 1957 on Mauna Loa (Keeling et al., 1976)), to a global network of monitoring sites (Conway et al., 1994), and the recent satellite missions to measure $CO_2$ in the atmospheric column (Crisp et al., 2004; Buchwitz et al., 2005; Yokota et al., 2009). Direct measurements of the difference between the partial pressure gradient of $CO_2$ between sea water and the overlying air ($\Delta pCO_2$) have also been available since the early 1970s (Takahashi et al., 2009). These measurements enabled the study of variability in sinks and sources of $CO_2$ at seasonal to interannual time scales. However, most of these time-series are still short, hampering the study of variability on scales longer than a few decades. As direct high-precision $CO_2$ measurements are only available for the later decades of the 20th century, ice-core records remain a valuable source of information about atmospheric $CO_2$ variability and trends during earlier periods.

The high-resolution measurements of $CO_2$ and the isotopic signature of carbon (usually expressed as $\delta^{13}C$) in air from ice-core and firn (unconsolidated snow) air from the Law Dome ice sheet in Antarctica (Etheridge et al., 1996; Francey et al., 1999; MacFarling Meure et al., 2006) encompass the last millennium, while extending to the present and overlapping with direct atmospheric observations. The Law Dome data remain unique for the period of the 1940s as the other recent high-resolution ice-core $CO_2$ record (from West Antarctic Ice Sheet Divide, WAIS-D) is restricted to years before 1940 due to contamination of gas records in ice collected from shallow depths (Ahn et al., 2012). The Law Dome record was used to study the variability at decadal scales in $CO_2$ sources and sinks during the 20th century (Joos and Bruno, 1998; Joos et al., 1999; Trudinger et al., 2002a). A conspicuous feature in the atmospheric $CO_2$ record is a stabilization of $CO_2$ concentration at around 310-312 ppm from 1940 to the early 1950s (Etheridge et al. (1996), Figure 1). This $CO_2$ stabilization was reconfirmed in the high density measurements from MacFarling Meure et al. (2006) and more recently by Rubino et al. (2013).

Assuming the estimates and their uncertainty range by Boden et al. (2009)are correct, the $CO_2$ plateau could not have been the result of $CO_2$ emissions from fossil-fuel burning and cement production going down to zero. The only available land-use change (LUC) emission estimates over the 20th century, based on observation-driven (bookkeeping) models, also do not report any decrease during this period (Houghton, 2003; Hansis et al., 2015). Brovkin et al. (2004), using an intermediate complexity Earth system model, found that, even in the absence of land-use emissions, the atmospheric $CO_2$ concentration should have risen by 2.6 ppm during the 1940 to 1950 period, a rate comparable to previous decades. Accounting for land use change scenarios, (Ramankutty and Foley, 1999; Klein Goldewijk et al., 2011) added 0.7–1 ppm to this $CO_2$ rise due to fossil fuel emissions. Simulated $\delta^{13}C$ in the atmosphere was in line with the Law Dome record during 1940s, but offset by -0.2‰after 1950, which might indicate overestimation of the land-use emissions after WW2.

Due to the smoothing of the short-term variations of atmospheric $CO_2$, the plateau in the ice-core record may be either due to a remarkably strong uptake of $CO_2$ lasting a few years, or a sustained uptake matching the anthropogenic emissions during the period, either by the land or ocean reservoirs (or a combination of both).



Double deconvolution of $CO_2$ and $\delta^{13}C$ suggests that this increased sink was dominated by ocean uptake (Joos et al., 1999; Trudinger et al., 2002a; Rubino et al., 2013). Etheridge et al. (1996) noticed that persistent El Niño sequences in 1895-1898 A.D. and 1911-1916 as well as the 1940s coincided with small decreases in the $CO_2$ growth rate, and Joos et al. (1999) hypothesized that the very strong El Niño event that lasted from 1940 until 1942 (Brönnimann et al., 2004) may have been

responsible for reduced upwelling of carbon-rich waters in the Eastern Pacific, causing an abnormal increase of the global ocean sink. However, this hypothesis remains controversial and, moreover, in spite of the high quality of the Law Dome $\delta^{13}C$ record, the scatter and uncertainty in the data are relatively high and they affect how well it is possible to partition the biospheric and oceanic fluxes. Errors in the $\delta^{13}C$ data may lead to spurious and highly correlated terrestrial and oceanic fluxes (Francey et al., 1995). The magnitude of uncertainties in the $\delta^{13}C$ ice-core measurements can be hard to estimate accurately, and the

choice of the uncertainty range may result in significant differences in the magnitude of the resulting fluxes.

Rafelski et al. (2009), using a single-deconvolution of the $CO_2$ record and a simple land-surface model, pointed to an increased terrestrial sink during the 1940s. This sink was related to change in temperature. Single deconvolutions do not use the $\delta^{13}C$ information, and when terrestrial uptake is used to explain the 1940s plateau they produce a peak in $\delta^{13}C$ that appears to be inconsistent with the ice core $\delta^{13}C$ measurements, although the differences are not large compared to the measurement

uncertainties (Trudinger et al., 2002a).

Furthermore, even if the unusually long 1940-42 El Niño did induce strong oceanic uptake, it is not clear that it should have led to a decrease in $CO_2$ growth rate, as El Niño periods in recent decades have usually been associated with a net increase in atmospheric $CO_2$ growth rate (Keeling et al., 1995; Ballantyne et al., 2012). The occurrence of El Niño leads to reduced outgassing of $CO_2$ in the tropical Pacific due to the slow-down of vertical upwelling of carbon and nutrient-rich subsurface

waters, driven by weaker trade winds (Feely et al., 2006). However, the magnitude of the ENSO impact on oceanic uptake differs significantly between studies, with approaches based on $\delta^{13}C$ analysis pointing to anomalies of 1.5–2.5 Pg C yr$^{-1}$ (Keeling et al., 1995; Joos et al., 1999; Trudinger et al., 2002a), while atmospheric $CO_2$-based methods point to anomalies of only 0.1–0.5 Pg C yr$^{-1}$ (Bousquet et al., 2000; Rödenbeck et al., 2003; Feely et al., 2006), consistent with the values obtained from observations of $\Delta pCO_2$ in the equatorial Pacific by Feely et al. (2006) or for the global ocean (Wanninkhof et al., 2013).

Furthermore, the enhancement of the global ocean sink during an El Niño event is usually offset by a much larger terrestrial $CO_2$ release, due to the response of land ecosystems to widespread drought conditions in the tropics (Sarmiento et al., 2010; Le Quéré et al., 2013) and increased fire emissions (van der Werf et al., 2004).

Here, we evaluate whether it is possible to reproduce the stabilization in atmospheric $CO_2$ during the 1940s using model-based records of sources and sinks for the 20th century and identify possible mechanisms to explain the plateau. We first

compare the atmospheric $CO_2$ growth-rate reconstructed using these datasets with the ice-core record to test their ability to capture the plateau. Additionally, we evaluate whether the ocean response to the 1940-42 El Niño may explain the atmospheric $CO_2$ stabilization. Finally, we analyse the response of the land-sink to this event using land-surface process models, and, given that land-use data are highly uncertain, test the possible contribution of LUC to explain the additional sink required to match observations.



## 2  Methods

### 2.1  Atmospheric $CO_2$

The Law Dome ice-core and firn air records of atmospheric composition were constructed by analysis of air trapped in imper-meable ice-cores or in firn layers at 4 sites on Law Dome in Antarctica (DE08, DE08-2, DSS and DSSW20K). They extend

back about 2000 years before present with very high air-age resolution and measurement precision. It is the only $CO_2$ record covering the 1940s and 1950s period and overlapping with the start of direct measurements (Etheridge et al., 1996; Francey et al., 1999; MacFarling Meure et al., 2006).

   Here, we use the atmospheric $CO_2$ concentration data from air samples from the DE08, DE08-2 and DSS ice cores, and from firn air from the DSSW20K record (Trudinger et al., 2002b) and from the South Pole from Rubino et al. (2013), shown

in Fig. 1. To compile this dataset, Rubino et al. (2013) added new $CO_2$ and $\delta^{13}C$ measurements to the record and revised sampling methods, uncertainty estimates and gravitational and diffusive mixing corrections relative to older measurements (e.g. Etheridge et al. (1996); Francey et al. (1999); MacFarling Meure et al. (2006)).

   The enclosure of air pores in ice is a gradual process that occurs in the lock-in zone, the transition layer between firn and impermeable ice. Due to the porosity of firn, there is also mixing between different parcels of air in the overlying layers.

Therefore, the composition of air in ice-core samples does not correspond to a single discrete year in the past, but rather to a mix of air parcels with different ages.

   The air age distribution, i.e., the temporal range of real world atmospheric composition sampled by each single ice-core measurement, can be quantified by a model of the processes (Trudinger et al., 2002b). Law Dome ice has an average spectral width (a measure of the spread of the distribution, Trudinger et al. (2002b)) of 4.5 years in DE08 and DE08-2, 5.8 years in DSS

ice samples and 7.0 years in DSSW20K firn samples (Trudinger et al., 2013; Rubino et al., 2013). More information about the characteristics of the original dataset may be found in Rubino et al. (2013).

   In order to derive a continuous time series of annual values from the individual measurements, we fit a smoothing spline curve to the ice-core measurements following the procedure described by Enting et al. (2006) which allows estimation of uncertainties in the spline, as well as in its derivative. The derivative corresponds to the annual atmospheric $CO_2$ growth rate

(hereafter AGR) (Fig. 1).

   However, when fitting a spline to data, a set of parameters needs to be chosen (regularization parameter and smoothing weights), which may affect the resulting spline (Enting et al., 2006). We performed sensitivity tests on the choice of these parameters (Fig. S1). While different choices of parameters and weights lead to very similar values of atmospheric $CO_2$ during the 20th century (Fig. S1, top), the $CO_2$ growth rate has much higher sensitivity to the different choices (Fig. S1, bottom). Here

we use $\lambda=30$ which results in 50% attenuation of variations shorter than 23 years, comparable to the 20 years of MacFarling Meure et al. (2006), and use unit weights for the fit, as in the standard definition of smoothing-spline (Enting et al., 2006), although the latter result in higher uncertainty relative to other choices of weights for the fit.

   A running piecewise trend adjustment was performed on the spline data between 1930 and 1960 (Fig. S2) to identify break points and the existence of a monotonically increasing or decreasing trend of atmospheric $CO_2$ for each trend section is tested





by a Mann-Kendall test. The fit with the smallest root mean square error of the adjustment was selected and the corresponding breakpoints defined as limits of the plateau. The resulting period spans from 1940 to 1950, as highlighted in Fig. 1 by the two vertical lines. During this period we found no significant trend in atmospheric $CO_2$. Note that the ice-core record is a smoothed and slightly shifted representation of the real atmospheric variations, and therefore the sink anomaly is more likely to be expected a few years after the stabilization becomes evident in the ice-core record (Trudinger et al., 2002b). Previous analysis of the plateau has varied by a few years in the timing of the maximum uptake. Joos et al. (1999) predicted maximum uptake in 1943, Trudinger et al. (2002a) in 1942 without consideration of the age distribution, and mid-1940s when age distribution is considered, in line with Rubino et al. (2013), that estimate that the event likely occurred some five years later than indicated in the ice-core record.

The growth rate of atmospheric $CO_2$ ($dC_{ATM}/dt$) corresponds to the net balance between anthropogenic emissions and the ocean and terrestrial fluxes:

$$\frac{dC_{ATM}}{dt} = AGR = E_{FF} + E_{LUC} - O - L \qquad (1)$$

with $E_{FF}$ being the anthropogenic emissions of fossil fuel burning and cement production, $E_{LUC}$ the net $CO_2$ emissions from changes in land-use, and O and L the ocean and land sink strength respectively. The total flux from the terrestrial biosphere is given by:

$$B = L - E_{LUC} \qquad (2)$$

We define here the emission terms $E_{FF}$ and $E_{LUC}$ as positive fluxes into the atmosphere, and the sink terms O, L, and B as positive fluxes out of the atmosphere.

## 2.2 Anthropogenic $CO_2$ emissions

### 2.2.1 Fossil fuel combustion and cement production

The Carbon Dioxide Information Analysis Center (CDIAC) provides annual estimates of $CO_2$ emissions from fossil fuel burning, cement production and gas flaring ($E_{FF}$) from 1751 to the present (Boden et al., 2009; Le Quéré et al., 2013). Here we use their most recent global estimates between 1900 and 2000, which have an uncertainty of $\pm 5\%$.

### 2.2.2 Emissions from Land-Use Change

The net $CO_2$ emissions from changes in land-use ($E_{LUC}$) are usually derived from information about changes in carbon stocks from cropland cultivation or pasture expansion and abandonment, wood harvest, shifting cultivation, deforestation/afforestation and forest regrowth after land abandonment. As this net flux cannot be directly measured, it is usually estimated using models that track carbon stocks in the different pools from inventories and historical accounts (the bookkeeping approach), or by





process-based models which simulate carbon fluxes due to imposed changes in photosynthesis and decomposition processes. It is important to distinguish between reconstructions of $CO_2$ fluxes based on gross changes in land use, and ones based on net changes, since the latter were found to underestimate fluxes by more than 0.5Pg C yr$^{-1}$ (Wilkenskjeld et al., 2014). Here, we use data from two bookkeeping methods that rely on gross land-use transitions: the bookkeeping datasets from Houghton

(2003) and from the "Bookkeeping of Land Use Emissions" (BLUE) model described by Hansis et al. (2015). We also use LUC emissions estimated by a set of process-based models, described in Sect. 2.3.4. This is intended to account for the loss of additional sink capacity, as discussed by Pongratz et al. (2014).

The bookkeeping model of Houghton (2003) is the one used in the Global Carbon Budget assessment (Le Quéré et al., 2013), and covers the period 1850-2005. It is mainly based on regional statistics from the Food and Agricultural Organization

(FAO) (Food and Agricultural Organization (FAO), 2010) and includes the effect of peat fires (from 1997 onwards) and fire suppression, the latter only for the USA. The model by Houghton (2003), allocates pasture preferentially to grassland, which may yield lower $CO_2$ emissions by reducing deforestation (Reick et al., 2010).

The BLUE model relies on the land use transitions from Hurtt et al. (2011) (which is based on the HYDE database (Klein Goldewijk et al., 2011) for cropland and pasture areas) to reconstruct fluxes between 1501 and 2012 in a spatially

explicit way. New cropland and new grassland are both taken proportionally from natural vegetation types. Two subsets of $E_{LUC}$ are calculated, one using vegetation and soil carbon stocks from Houghton et al. (1983), and the other using the modifications proposed by Reick et al. (2010), that feature generally lower carbon densities for natural vegetation and lead to lower emissions. More details about the data sources and methods can be found in the original literature.

## 2.3 Ocean and land sinks

Observation-based estimates of $CO_2$ exchanges between the atmosphere, the ocean and terrestrial ecosystems are only available since the 1970s (Takahashi et al., 1997; Baldocchi et al., 2001; Manning and Keeling, 2006; Peylin et al., 2013). Here, we use different reconstructions of the ocean and terrestrial sinks for the 20th century, based on indirect methods. The goal of this procedure is two-fold: to test the ability of these reconstructions to close the $CO_2$ budget and to gain insight into the drivers of the 1940s plateau.

### 2.3.1 Double-deconvolution of $CO_2$ and $\delta^{13}C$ records

Joos et al. (1999) used a double-deconvolution technique to reconstruct land and ocean fluxes from measurements of atmospheric $CO_2$ and $\delta^{13}C$ taken from the Law Dome ice-core record, between 1800 and 1990. Their analysis relied on a previous dataset (Etheridge et al., 1996; Francey et al., 1999) of the same $CO_2$ ice-core record used here (Rubino et al., 2013) to solve two mass-balance equations for atmospheric $CO_2$ and $\delta^{13}C$. The method uses prescribed carbon fossil-fuel emissions and their

$\delta^{13}C$ signature with a box-model to simulate isotopic disequilibrium fluxes between the atmosphere, ocean and biosphere (i.e. $F_B$), $O_J$ and $B_J$ respectively, shown in Fig. 2 (top panel). Trudinger et al. (2002a) used the same measurements in a Kalman filter double deconvolution. They came to the generally similar conclusion, namely that the oceans played a significant role in creating the 1940s plateau. These two double deconvolutions have some common weaknesses. Neither calculation considers





climate-driven variations in terrestrial isotopic discrimination (Randerson et al., 2002; Scholze et al., 2003), which likely co-vary with $CO_2$ fluxes that are also driven by climate. The calculations also do not consider changes in the distribution of C3 and C4 plants with time (Scholze et al 2008). Both of these effects may be important. It is possible to calculate both effects with process models, generally as part of a forward model calculation, but it would be problematic to calculate them in an inversion such as a double deconvolution. As with any inversion, the results depend on the choice of statistics such as the magnitude of uncertainties (Trudinger et al., 2002a) and the degree of smoothing of the fit to the ice-core measurements used in the mass balance method (Joos et al., 1999). In both cases, these choices define how much of the variability in the ice core is considered as 'signal' to be interpreted, and how much is considered 'noise' to be ignored. Such choices can be subjective, and lead to differences in the magnitudes of variations. The scatter in the Law Dome $\delta^{13}$C ice-core measurements at the time of the plateau is significant compared to the signal that we need to interpret to understand the cause of the plateau. Furthermore, emissions from LUC have the same isotopic signature as L, making it impossible to disentangle the two terms, and fluxes from the C4 photosynthesis pathway (which have a lower affinity for the lighter carbon isotopes) may be attributed to the ocean. Nevertheless, double deconvolutions interpret measurements that represent globally-aggregated signals, allowing estimation of the main decadal variability patterns in the land and ocean sinks due to changes in climate over long time-scales (Joos and Bruno, 1998; Joos et al., 1999; Trudinger et al., 2002a). These double deconvolutions may thus be used to compare with the patterns found in our model-based reconstructions.

### 2.3.2 Reconstruction of anthropogenic $CO_2$ uptake by the ocean

Several methods have been developed to estimate ocean $CO_2$ fluxes from observations (Takahashi et al., 1997; Rödenbeck, 2005; Manning and Keeling, 2006; Landschützer et al., 2015; Le Quéré et al., 2015); however, most of them cover only the last three decades of the 20th century.

Khatiwala et al. (2009) used an inverse technique to reconstruct the oceanic response to the anthropogenic perturbation, i.e., the uptake of anthropogenic $CO_2$ by the global oceans between 1765 and 2008. Their estimates of oceanic $CO_2$ uptake (henceforth $O_K$) and their respective uncertainties are shown in Fig. 3. In their reconstruction, the transport of anthropogenic $CO_2$ in the ocean is described by an impulse response function, using a kernel that describes ocean circulation and allows us to trace the transport of $CO_2$ from the surface to the deep ocean. This kernel is calculated from observations in recent decades of active and passive tracers: temperature, salinity, oxygen, naturally-occuring $^{14}$C, CFCs, and $PO_4$. However, in their approach ocean-circulation does not include natural variability, apart from a seasonal-cycle. Nevertheless, their reconstruction is the one used in most of the 20th century reconstructions (IPCC, 2013) and, despite not representing interannual to decadal variability, it sets a reference level about which we can define the range of ocean variability required to explain the plateau.

### 2.3.3 Ocean sink from CMIP5 models

Currently, analysis of the role of ENSO in variations in oceanic sink reconstructions from ocean general circulation models including biogeochemistry and driven by climate and atmospheric $CO_2$ observations is only available for the second half of the 20th century. This is due to the lack of atmospheric reanalysis for the early 20th century (Wanninkhof et al., 2013; Le Quéré



et al., 2015). One way of gaining insight into the possible role of the ocean in explaining the plateau could come from the analysis of coupled climate-carbon simulations over the 20th century. Despite the fact that the simulated variability is not necessarily in phase with the observed one, these simulations offer the opportunity to estimate the potential amplitude and patterns of carbon flux variability at interannual to decadal time-scales.

We evaluate the ranges of natural variability in the global ocean sink using outputs of global ocean $CO_2$ flux from a set of sixteen general circulation and Earth system models (GCMs and ESMs, respectively) used for the Coupled Model Inter-comparison Project Phase 5 (CMIP5), over the period 1860-2000. In order to match the time-scales of the ice-core record, the annual values of ocean fluxes were filtered according to the air-age distribution for $CO_2$ in DE08 ice (Trudinger et al., 2003) and anomalies are calculated as the departure from the 30-yr moving average.

Some of the models differ only in their atmospheric resolution or the representation of certain physical processes in the ocean, whose details are given by Anav et al. (2013). In the historical simulation, increasing atmospheric $CO_2$ concentrations were prescribed, as well as external forcings such as sulphate aerosols, solar radiation variability, and volcanic eruptions. When considering only one realization of each model, the internal climate variability patterns and their influence in the resulting outputs may not be fully captured (Deser et al., 2012). Therefore, we also evaluated global ocean flux outputs from simulations

using the same forcings as those mentioned above, but initialized with perturbed initial conditions. The IPSL-CMA5 performed a set of six realizations of the historical simulation from the IPSL-CMA5-LR, plus three realizations from IPSLCMA5-MR, which may provide a better depiction of the ranges of natural variability to be expected in the ocean sink. For these simulations we also analyse variability in tropical sea-surface temperature, which allows evaluation of the contribution of ENSO to the strongest anomalies in oceanic uptake.

### 2.3.4   Land sink from DGVMs

The land sink may be reconstructed with a dynamic global vegetation model (DGVM), forced with climate observations and atmospheric $CO_2$ from ice-core data, as performed in the TRENDY project (Sitch et al., 2013), and used in other reconstructions of the $CO_2$ budget (Le Quéré et al., 2015). These models simulate water and carbon exchanges at the ecosystem level, and some models also simulate vegetation dynamics, disturbance and nutrient limitation (Table 1).

In experiment S2 from TRENDY, models are forced with climate observations from the CRU-NCEP v4 between 1900 and 2000, but do not represent land-use change. Monthly Net Biome Production fields from each model were integrated globally and aggregated over each year, to produce an annual time-series for the 20th century. Figure 3 shows the annual values of the global land sink as evaluated by the group of DGVMs ($L_{DGVM}$).

### 2.4   Closing the $CO_2$ budget

As discussed above, these different sets of data for the carbon budget terms should, if correct, allow reconstruction of the Law Dome $CO_2$ record during the 20th century. The estimates of Joos et al. (1999) were originally calculated from earlier





measurements of the Law Dome record, which were confirmed by new measurements. Therefore, atmospheric $CO_2$ growth-rate calculated from fossil-fuel emissions and their ocean and biospheric fluxes ($AGR_J$), i.e.:

$$AGR_J = E_{FF} - O_J - B_J \qquad (3)$$

should be similar to the AGR record resulting from the value obtained from our spline-fit on atmospheric $CO_2$ concentration.
However, it should be noted that in Joos et al. (1999) the smoothing is stronger. The values of $O_J$ and $B_J$ are shown in Fig. 2 together with the resulting $AGR_J$ between 1900-1990 and the corresponding difference with the observations (i.e. $AGR_J$ minus AGR, $\Delta AGR_J$).

The other sets of data, being calculated using very different techniques, are largely independent from the $CO_2$ record and from each other. However, it should be noted that in reality these fluxes are not entirely independent from each other. For
example, the emissions resulting from LUC will depend to a certain extent on the carbon stocks of the terrestrial ecosystems, i.e., in previous states of L. This is partially taken into account in DGVMs forced with LUC, but not in the other datasets. The resulting $CO_2$ budget using the different datasets for each term may be calculated using Equation 1, using $E_{FF}$ data and respective range, the ocean uptake reconstruction from Khatiwala et al. (2009) ($O_K$), the land-sink from DGVMs ($L_{DGVM}$) and the four $E_{LUC}$ estimates ie.:

$$AGR_i = E_{FF} + E_{LUC-i} - O_K - L_{DGVM}; \qquad (4)$$

With $i = (H; B; Blc; DGVM)$, referring to each of the four datasets used to estimate emissions from land-use change (Houghton, BLUE, BLUE with low C-stocks, and DGVMs, respectively) and $L_{DGVM}$ refers to the inter-model median of the global land sink from DGVMs (S2 experiment), shown in Table 2. To be compared with AGR from the ice core, the annual values of AGR computed using 4 need to be smoothed in order to match the air-age distribution of $CO_2$ in air trapped in
ice-bubbles at DE08 as proposed by Trudinger et al. (2003).

It should be noted that the AGR in Eq. 4 suffers from inconsistencies between terrestrial emission and sink terms when $E_{LUC}$ is derived from bookkeeping rather than DGVM models: while $L_{DGVM}$ includes the effects of changing environmental conditions, which historically created a sink, the bookkeeping estimates assume that carbon densities do not change over time, but keep them fixed at (the higher) observational values from recent decades (Houghton et al., 1983). This creates a tendency
towards overestimating early land-use emissions, likely some 10% for the industrial era (Stocker and Joos, 2015). Furthermore, the bookkeeping estimates do not include the loss of additional sink capacity (Gitz and Ciais, 2003; Strassmann et al., 2008; Pongratz et al., 2014). DGVMs do include the loss of additional sink capacity in their $E_{LUC}$ by using the S2 experiment of no land-use change under transiently changing environmental conditions as reference, so that the loss of the increased carbon stocks of forests that are replaced by agriculture are attributed to $E_{LUC}$. While the effect of constant carbon densities in the
bookkeeping method leads to AGR being overestimated for earlier decades, the effect of replaced sinks leads to AGR being underestimated. However, this effect becomes significant only with the strong climate change after the 1950s (Stocker and Joos, 2015).





The CO$_2$ growth rate during the 20th century, calculated from each set of data, is shown in Fig. 3 with the corresponding departure from the observed values ($\Delta$AGR$_i$). We represent the uncertainty range of the reconstructions as the uncertainty in E$_{FF}$ ($\pm$5%), O$_K$ (reported by Khatiwala et al. (2009)) and each individual E$_{LUC}$ estimate ($\pm$0.5 PgC uncertainty estimated by Houghton et al. (2012) for the bookkeeping models, and model spread for the DGVM).

The differences between each reconstruction and observations during the period 1940-1950 are summarized in Table 3 and provide an estimate of a residual sink further required to explain the CO$_2$ stabilization.

## 2.5 Testing LUC with idealized experiments

The LUC component of the carbon budget is one of the most uncertain terms (Houghton et al., 2012; Gasser and Ciais, 2013; Pongratz et al., 2014), and is as large as E$_{FF}$ in the first part of the 20th century. The $\delta^{13}$C record provides a constraint on the

relative contribution of the oceanic and terrestrial fluxes to the observed CO$_2$ emissions. This allows evaluation of the extent to which land-use change processes could contribute to the residual CO$_2$ sink. Here, we perform a set of idealized experiments to estimate the contribution of different terms of LUC to the overall carbon balance, as well as their compatibility with the $\delta^{13}$C record.

We use an updated version of the relatively simple coupled carbon-cycle and climate model OSCAR (Gasser and Ciais,

2013) to integrate the different components of the carbon-budget in a realistic mathematical and physical framework. The model includes a mixed-layer impulse response function representation of the ocean carbon-cycle (Joos et al., 1996). Carbonate oceanic chemistry is sensitive to atmospheric CO$_2$ and temperature change, and stratification is accounted for by changing the mixed-layer depth according to sea-surface temperature change, following CMIP5 models. The pre-industrial land carbon pools and fluxes are calibrated on the multi-model average of the TRENDY v2 models (Sitch et al., 2013). Net Primary Production

(NPP) then responds to varying CO$_2$ and climate, and heterotrophic respiration to varying climate, all of which are calibrated on CMIP5 models. OSCAR embeds a bookkeeping module (Gitz and Ciais, 2003) capable of calculating its own CO$_2$ emissions from land-use change, on the basis of land-cover change, wood harvest and shifting cultivation area inputs. Land-use change information is aggregated in ten different regions from the original dataset from Hurtt et al. (2011).

The variation in the stable carbon isotopic composition ($\delta^{13}$C) may be calculated from the balance of the different CO$_2$

fluxes (Tans et al., 1993; Hellevang and Aagaard, 2015) which are simulated by OSCAR using:

$$\delta^{13}C^t - \delta^{13}C^{t-1} = C^{t-1} \times \left\{ \delta_f^t E_{FF}^{t-1} + \delta_o^{t-1} OA^t + \delta_{lb}^{t-1} F_B^t - (\delta^{13}C^{t-1} + \epsilon_o)AO - (\delta^{13}C^{t-1} + \epsilon_{lb})NPP \right\} \times \Delta t \qquad (5)$$

where $t$ refers to time, C is the atmospheric CO$_2$ concentration, E$_{FF}$ denotes fossil fuel emissions, OA and AO the gross ocean-atmosphere and atmosphere-ocean fluxes respectively, F$_B$ is the gross flux between terrestrial ecosystems and the atmosphere (i.e. emissions from heterotrophic respiration, mortality, fires and land-use change), and NPP is the global net primary

production. In OSCAR all these fluxes (in Pg C yr$^{-1}$) are calculated as variations (e.g. $\Delta$NPP) from an initial state in 1700 (NPP$_0$), whose values are given in Table 4, together with the values used for the fractionation ratios and isotopic composition of the different reservoirs.





The standard set-up of the OSCAR model does not capture the stall in atmospheric $CO_2$ during the 1940s, despite performing relatively well during most of the 20th century. This failure may be due to a variety of reasons, as discussed by Gasser (2014). Nevertheless, it allows us to track the individual contribution of each budget term to the overall $CO_2$ budget. Here, the OSCAR model is used to evaluate the relative effect of hypothetical extreme land-use changes during 1940-1950 in AGR for

instance related with the abrupt socioeconomic changes imposed by WW2 (and prolonged during the early post-war period) in many regions. Our idealized experiments exaggerate the magnitude of the hypothetical LUC during 1940-1950, but they allow quantification of their relative impact on the resulting AGR and $\delta^{13}C$, as compared with the standard OSCAR set up, providing an indication of how much a given LUC transition may contribute to the global carbon balance during the period 1940-1950.

    The global area under LUC transitions during 1940-1950 used in the default OSCAR set up is shown in Table 5. On the first

test (T1), we set all the transitions from forest to other land-cover types to zero between 1940 and 1950, i.e., artificially and abruptly stopping deforestation over the globe in 1940. The second test (T2) doubles the area corresponding to forest expansion in each year (T2). The third test prescribes a halt in all expansion of cropland and pasture areas (T3), which indirectly also sets deforestation to zero, since forest is only lost to either crop or pasture (Table 5); the last test is to stop all wood harvest (T4).

## 3 Results

### 3.1 Reconstructions of $CO_2$ sources and sinks

The record of emissions from fossil fuel and cement production during the 20th century (Fig. 2) show a slow increase of $E_{FF}$ at a rate of ca. 0.02 Pg C yr$^{-2}$ during the first four decades, punctuated with periods of slight decrease. From 1940 to 1950, $E_{FF}$ was on average 1.4 Pg C yr$^{-1}$ and, in spite of a small decrease during 1945-1946, even accelerated, with a rate of change of 0.05 Pg C yr$^{-2}$ during the full period. As uncertainty in $E_{FF}$ is also very small ($<\pm0.1$ Pg C yr$^{-1}$) in the first half

of the 20$^{th}$ century, and a stabilization of $CO_2$ would imply $E_{FF}$ being zero for the whole period, its role in explaining the $CO_2$ stabilization in the 1940s is excluded. Here we evaluate whether the available sources of data about the other terms of Eq. 1 allow reconstruction of the plateau.

    Given that the estimates of ocean and biospheric fluxes from Joos et al. (1999), $O_J$ and $B_J$, were calculated using a previous version of the $CO_2$ record used here, they are expected to reproduce the observed variations in $CO_2$, as given by the general

agreement between observations (AGR) and reconstructed (AGR$_J$) shown in Fig. 2 (top panel). However, in spite of the uncertainty limits of AGR$_J$ encompassing the observations, discrepancies are found for some periods of the century, as the one from 1940-1950, and the one from the late 1960s until 1980. These are more evident when analysing the difference between reconstruction based on Joos et al. (1999) and observations, $\Delta$AGR$_J$ (Fig. 2, bottom panel). This is likely due to the different degrees of smoothing used in Joos et al. (1999) and here, as exemplified in Fig. S1.

The reconstructions performed using the different $E_{LUC}$ estimates, the ocean sink from Khatiwala et al. (2009), $O_K$, and the terrestrial uptake from DGVMs ($L_{DGVM}$) are shown in Fig. 3 (bottom panel). The discrepancies present variability patterns with different time scales, with a deviation from zero beginning around 1910, increasing up to a maximum ca. 1950 then decreasing back to zero around 1990 and a decadal variability about this longer term variation. All reconstructions overestimate AGR





between 1940 and the mid-1970s, and this overestimation is particularly large around 1945 and 1960. Results from $E_{LUC-H}$ and $E_{LUC-DGVM}$ are similar during most of the century and lead, generally, to lower discrepancies between reconstructions and observations, as compared with $E_{LUC-B}$ data. In the case of the two BLUE datasets, AGR is overestimated during most of the 20$^{th}$ century, and by up to 1-2 Pg C yr$^{-1}$ in two periods: the 1940s and 1950s-60s.

During the plateau period (Table 3), the values of $\Delta$AGR are, as expected lower for AGR$_J$ than for the other datasets, although the absolute uncertainty of the reconstruction is one order of magnitude higher (1.4 Pg C yr$^{-1}$) than the estimated misfit ($\Delta$AGR, 0.1 Pg C yr$^{-1}$ ), i.e., the extra sink required to match observed AGR. In AGR$_J$, the uncertainty is likely overestimated because in the double deconvolution, O$_J$ and B$_J$ have anti-correlated errors. If this had been taken into account, uncertainty would be similar to the one in their $CO_2$ observations, while it is generally of the same magnitude as the estimated values, still increasing from 1960 onwards.

For the independent datasets, the mismatch with observations is smaller for $E_{LUC-H}$ and $E_{LUC-DGVM}$ (0.9 and 1.2 Pg C yr$^{-1}$, respectively), reaching 2.0 Pg C yr$^{-1}$ for $E_{LUC-B}$. Part of the discrepancy observed during the plateau period in AGR estimated using Houghton and BLUE datasets results from the consistently higher values of BLUE over the whole century. The relative variation in $\Delta$AGR$_B$ and $\Delta$AGR$_{Blc}$ during 1940-1950 relative to the 1920s and 30s roughly matches the one observed in the other two datasets.

However, it should be noted that DGVMs also differ considerably in their estimates of the land sink during 1940-1950, with one model (LPJmL) even estimating a terrestrial source rather than a sink during the period (Table 2).

## 3.2 Testing the hypothesis for the plateau

As shown previously, the datasets of carbon budget terms ($E_{FF}$, O$_K$, L$_{DGVM}$ and the different $E_{LUC}$ data) lead to overestimation of AGR by 0.9 to 2.0 Pg C yr$^{-1}$, for $E_{LUC-H}$ and $E_{LUC-B}$ respectively. Thus, there is a sink missing in the budget, that could be explained by: i) decadal variability in the ocean sink not represented in O$_K$; ii) processes absent from all the TRENDY models causing extra land uptake in ecosystems without land-use change iii) land-use change processes that lead to carbon uptake and are not, or not sufficiently, included in the current datasets.

### 3.2.1 Ocean variability

Despite both ocean reconstructions (O$_J$ and O$_K$) generally agreeing on the long term trend of the ocean sink (Figs. 2 and 3), O$_K$ presents a smooth increase, consistent with the evolution of atmospheric $CO_2$, while O$_J$ points to the existence of large multi-decadal variations superimposed on the increasing trend, the largest of them coinciding with the plateau period. During 1940-1950 the two datasets differ by about 0.5 Pg C yr$^{-1}$ (O$_K$ ca. 0.7 Pg C yr$^{-1}$ and O$_J$ ca. 1.2 Pg C yr$^{-1}$ on average), providing a reference value for the possible contribution of natural variability in the ocean to the sink required to stabilize atmospheric $CO_2$.

It is important to evaluate whether an enhancement of the ocean sink of the magnitude reported by Joos et al. (1999) is likely to have occurred during the first half of the 20th century. Such reinforcement of oceanic $CO_2$ uptake could only be explained by natural variability, as given by the large difference between O$_J$ and O$_K$ for this period (0.5 Pg C yr$^{-1}$), for instance due to a





strong El Niño event, as suggested by previous works (Joos et al., 1999; Trudinger et al., 2002a) for the exceptional 1940-42 El Niño (Brönnimann et al., 2004).

This may be tested by evaluating the variability patterns of $CO_2$ fluxes in the global ocean calculated by the set of 16 ESMs from CMIP5 for the historical period. Although the models are not expected to reproduce the exact temporal evolution of the

ocean sink because they simulate their own climate variability, it is possible to test their ability to represent decadal departures of magnitude of 0.5 Pg C yr$^{-1}$ from the long-term trend (as the difference between $O_K$ and $O_J$) or up to 2.0 Pg C yr$^{-1}$ (if we consider the residual sink to be in the ocean).

The anomalies of the ocean sink calculated by the models for the historical simulation (prescribed atmospheric $CO_2$ and external forcings), filtered to match the ice-core air-age distribution are shown in Fig. 4. As data are smoothed, the anomalies

correspond to a long-term pattern, rather than an annual anomaly. The variation ranges estimated by the models are about half of the ones suggested by $O_J$, with most anomalies being smaller than $\pm 0.15$ Pg C yr$^{-1}$, although in some models (e.g. GISS-E2-R-CC and IPSL-CM5A-MR) anomalies may reach values of about $\pm 0.2$ Pg C yr$^{-1}$. The anomalies in ocean $CO_2$ uptake present multi-decadal variations which are consistent between the 16 models and are due to the ocean response to the $CO_2$ forcing. In particular, during the plateau period, most models estimate lower ocean uptake because of the slow-down of the

anthropogenic perturbation. The inter-model comparison indicates that, assuming the magnitudes of variability of the modelled ocean fluxes are representative of the real ocean, an anomaly of more than ca. 0.2 Pg C yr$^{-1}$ in the ocean sink is unlikely to be registered by the ice-core record.

Nevertheless, to account for the impact of natural variability in ocean fluxes, it is advisable to consider a larger number of realizations for each model, given that results may differ considerably, especially in the time-scales of interest to this study

(Deser et al., 2012). The global ocean $CO_2$ uptake estimated by six realisations from IPSL-CMA5-LR and the three from IPSL-CM5-MR is shown in Figure 5a. Some of the different simulations reveal strong decadal variations, with anomalies varying (in some cases) by $\pm 0.3$ Pg C yr$^{-1}$. These variations are more pronounced for the model with higher spatial resolution (IPSL-CMA5-MR), suggesting a possible influence of smaller scale processes that control internal variability of the ocean, for instance better representation of the westerlies in the Southern Ocean (Hourdin et al., 2012). Nevertheless, such a range of

variation is consistent with observation-based estimates for the late 20th century (Landschützer et al., 2015).

The strongest positive anomalies in the ocean sink for each of the IPSL-CMA5 simulations, and the corresponding peak year are presented in Table 6, together with the corresponding variations in the east tropical Pacific sea-surface temperature (SST) (Fig. 5b and Table 6). Only three out of the nine simulations present strong ocean uptake coincident with warming (but very feeble) of the tropical oceans: *r1,r4* and *r6* from IPSL-CMA5-LR. This is consistent with the reduced upward transport of

carbon-rich water from the deep ocean, associated with weaker upwelling due to the persistence of warmer surface temperatures (as during El Niño events). However, it is not possible to establish a straightforward link between tropical Pacific SST and the enhancement of the ocean sink for any of the other simulations.





### 3.2.2 Land response to climate

If the additional sink were provided by land, and considering the inter-model median $L_{DGVM}$ of 0.8 Pg C yr$^{-1}$, a total terrestrial uptake of more than 1.5 Pg C yr$^{-1}$ would be needed. This magnitude is comparable to the average land sink in the early 2000s (1.3 Pg C yr$^{-1}$) estimated by atmospheric inversions (Peylin et al., 2013), when the effects of $CO_2$ fertilization are already
important (Friedlingstein et al., 2006).

It is thus worth testing whether DGVMs capture realistically the response of terrestrial ecosystems to the climate forcing during the plateau period, as well as to the strong El Niño event (1940-42), as ENSO impacts on regional climate and terrestrial ecosystems have been studied for later events (Diaz et al., 2001; Bastos et al., 2013) and therefore provide a known reference to analyse the expected anomalies in the climate forcing and the corresponding simulated response.

Table 2 shows the average land sink estimated by DGVMs during 1940-1950 and the 1940-42 El Niño. DGVMs estimate in general a relatively strong terrestrial sink during the plateau, except LPJmL which simulates a 0.43 Pg C yr$^{-1}$ terrestrial source during the period. When compared to the period 1900-1930, all DGVMs estimate an increased sink in the Northern Hemisphere, especially at high latitudes, coinciding with generally warmer and wetter conditions throughout most of North America and Eurasia (Figure 6). This increased sink is mostly due to strong enhancement in gross primary productivity (Figure
S3), consistent with the increased growth observed in tree rings in the Northern Hemisphere (Briffa et al., 1998). In the tropics, models diverge significantly in the anomalies in $CO_2$ uptake in response to the temperature (generally lower) and precipitation (above average in most regions) patterns during the plateau. Differences in model sensitivity to temperature and precipitation, or lack of proper fire representation, may explain part of this mismatch.

Five of the nine models indicate a reduction of terrestrial uptake in 1940-42 (as compared to the plateau period), expected
during a warm ENSO event although not as strong as the response of the terrestrial sink to El Niño registered in the late 2000s (Sarmiento et al., 2010; Le Quéré et al., 2013). In general, temperature anomalies (Figure S4, left panel) over land in 1940-42 present an El Niño-like distribution (Diaz et al., 2001; Brönnimann et al., 2007), with warming in most of the tropical and sub-tropical regions, and the strong cooling over Europe reported by Brönnimann et al. (2004). However, dry conditions during 1940-42 in the forcing are confined to part of northern South America and the Philippines, rather than the characteristic overall
drying of part of Amazonia and sub-tropical South America, southern Africa or Australia that usually leads to weaker $CO_2$ uptake by land ecosystems during positive ENSO events (Diaz et al., 2001; Bastos et al., 2013).

Although most models capture the reduction in terrestrial uptake in the tropical regions (Fig. 6), some estimates of tropical anomalies are very small. At the same time, most models estimate a strong enhancement of the sink in northern latitudes, especially in the Eurasian region, which partially offsets the small decrease of $CO_2$ uptake in the tropics. The enhanced northern
$CO_2$ uptake during El Niño derives from a combination of high photosynthesis in North America (where strong warming is registered) and a combination of enhanced photosynthesis and low respiration in Europe (which registers negative temperature anomalies during all seasons except summer). The very strong response to the latter effect in some models explains the very small land sink anomalies found for most of the models, and the enhanced sink identified by CLM4.5, JSBACH and LPJmL.





### 3.2.3 Land-use change

The differences in $E_{LUC}$ for the four extreme hypothetical scenarios and the standard OSCAR run are shown in Fig. 7 (top), as well as the comparison of the resulting changes in the atmospheric $CO_2$ growth-rate (center) and $\delta^{13}C$ (bottom) with the observational values. The average differences in LUC emissions and resulting AGR during 1940-1950 are summarized in Table 7.

The two largest reductions in $E_{LUC}$ result from halting either deforestation or crop and pasture expansion, which lead to an average reduction of 0.51 Pg C yr$^{-1}$ and 0.49 Pg C yr$^{-1}$ during the decade, peaking at about 0.8 in 1950, when the standard OSCAR LUC are resumed. The loss of forest to cropland and pasture influences the fluxes resulting from crop and pasture expansion, as shown by the small differences between the two emission trajectories (T1 and T3). Due to the interactions between these two transitions and the land sink, the resulting difference in atmospheric $CO_2$ is about 25% smaller, of 0.38 and 0.36 Pg C yr$^{-1}$, for T1 and T3 respectively.

Stopping wood harvest during 1940-1950 (T4) leads to $E_{LUC}$ 0.23 Pg C yr$^{-1}$ lower than the standard simulation, resulting in AGR differences of 0.17 Pg C yr$^{-1}$. However, in this case, $E_{LUC}$ increase rapidly from about 1950 onwards and even surpass the values estimated by the standard simulation, which may be related to the predominance of biomass burning and fast decomposition processes during the first years after resuming harvest. Although smaller in magnitude, a hypothetical doubling in the area under forest expansion (T2) leads to a decrease in $E_{LUC}$ of 0.07 Pg C yr$^{-1}$, and impacting AGR by 0.06 Pg C yr$^{-1}$.

The relative abundance of carbon isotopes $^{12}C$ and $^{13}C$ depends on the carbon fluxes between the different reservoirs, as the driving processes (e.g., photosynthesis, fires, respiration, ocean dissolution) have specific isotopic fractionation ratios (Tans et al., 1993). The isotopic signature of carbon in $CO_2$ samples (usually expressed as $\delta^{13}C$) thus provides a constraint on the relative contribution of each process to the observed variations in atmospheric $CO_2$ concentration. Isotopic data from the ice-core record reveal a flattening of $\delta^{13}C$ between c.a. 1915 and 1950 (Rubino et al., 2013).

The $\delta^{13}$ calculated using the standard OSCAR set-up generally remains within the uncertainty range of the observations, except during 1950-1960 and the late 1990s. In spite of performing rather well for most of the century, the standard set-up does not fully capture the flattening of the $\delta^{13}$ record during the 1915-1950 period.

An increase in $\delta^{13}$ during the plateau period is observed for all the idealized experiments, consistent with an increased terrestrial sink. Despite our tests imposing changes in $E_{LUC}$ only for the 1940-1950 period, differences between their $\delta^{13}C$ signature and the one from the standard set-up are noticeable until the late 1960s. Experiments T1 and T3 lead to a stronger increase in $\delta^{13}C$ relative to the standard simulation, but still remain roughly within the uncertainty limits of the observations between 1940 and 1950 and actually remain closer to observed $\delta^{13}C$ in the subsequent decade.

## 4 Discussion

We find that the datasets of anthropogenic $CO_2$ emissions combined with the reconstructions of carbon uptake by terrestrial ecosystems and the ocean are not able to reproduce the decrease in atmospheric $CO_2$ growth rate between 1940-1950 registered





in observations. A further sink of at least 0.9 Pg C yr$^{-1}$ is still required. While uncertainty in emissions from fossil fuels is much smaller than the sink required, uncertainty in the other terms is very high.

### 4.1 CO$_2$ sinks during the plateau

An ocean sink of about 1.2 Pg C yr$^{-1}$ during the 1940s, as in the Joos et al. (1999) dataset, is needed to explain partly the

observed CO$_2$ plateau; such a sink is compatible with the occurrence of a strong ocean uptake anomaly due to natural climate variability superimposed on the anthropogenic perturbation trend. The variation range of different realizations of the IPSL-CMA5 model forced with perturbed initial conditions, is within the variability range found for ENSO impacts on oceanic CO$_2$ uptake in the late 20th century (0.1-0.5 Pg C yr$^{-1}$, (Bousquet et al., 2000; Rödenbeck et al., 2003)). Other works have suggested an ocean uptake of 2-2.5 Pg C yr$^{-1}$ during the 1940s (Trudinger et al., 2002a), or in response to later ENSO events

(Keeling et al., 1995), which appears to be too high in light of the variations simulated by the models and the recent estimates from atmospheric inversions.

The role of an extreme ocean uptake event as, for instance, in response to the 1940-42 El Niño, does not seem likely to have been the sole driver of the plateau – other sources of variability from the ocean may need to be considered. Resplandy et al. (2015) have analysed unforced natural variability in the ocean using century-long simulations from a set of six ESMs

(a sub-set of the ones we use in this study). At interannual to decadal time-scales, models indicate a strong contribution of the Southern Ocean to the global ocean sink, due to: (1) variations in wind-stress and deep-water upwelling controlled by the Southern Annular Mode (SAM), and (2) the occurrence of deep convective events that trigger a reduction in sea-ice coverage and intense mixing of surface waters with carbon-rich deep-waters. Regarding SAM-induced variability, the changes in atmospheric circulation in the late 2000s have been recently linked to a remarkable increase in CO$_2$ uptake by the Southern

Ocean, from about 0.6 Pg C yr$^{-1}$ in 2002 to 1.2 Pg C yr$^{-1}$ in 2011 (Landschützer et al., 2015). If variations of this order of magnitude in the Southern Ocean would be accompanied by non-cancelling anomalies in the tropical Pacific, one could expect a higher contribution of the ocean to the global sink during the 1940s. Regarding the convective events, climate models suggest a long multi-decadal time-scale, from 20-30 to 50-60 years depending on the model, which makes them relatively rare events even for a century-long record. Satellite observations indicate the existence of such a deep convective event in the 1970s

(Gordon, 1978), but there are no observations for the 1940s. Given the lack of observation-driven datasets able to capture these variabilities, this hypothesis remains speculative.

Considering a contribution of 0.5 Pg C yr$^{-1}$ due to natural variability in the ocean, as estimated by Joos et al. (1999) and recent observations, a further (terrestrial) sink of 0.4-1.5 Pg C yr$^{-1}$ is required. DGVMs used to characterize the land-sink during the 20th century indicate that terrestrial ecosystems constituted an important CO$_2$ sink, taking up about 0.8 Pg C yr$^{-1}$

during the period between 1940 and 1950 in response to generally warmer and wetter conditions. The models estimate a small decrease of the terrestrial sink during the strong El Niño event of 1940-42 (and even enhancement in some models), which is not fully consistent with the more recent observations of the terrestrial response to ENSO (Sarmiento et al., 2010; Le Quéré et al., 2013). Despite most models capturing a decrease in CO$_2$ uptake in the tropics during the El Niño in response to dryness, the reduction is likely underestimated, as DGVMs are known to have problems in representing fire disturbance





(Murray-Tortarolo et al., 2013). In any case, the aforementioned discrepancies would further reduce the terrestrial sink, rather than helping to explain the enhanced sink needed. On the other hand, the inconsistency of the land-sink response to El Niño with recent observations may also be due to the climate forcing, which does not represent the characteristic drying pattern over most of the Southern Hemisphere.

Before 1950, especially during WW2, the global meteorological network coverage was poor in comparison with the late 20th century. Moreover DGVM simulations rely on CRU-NCEP v4, which uses the CRU dataset for monthly data and NCEP/DOEII reanalysis to generate 6-hourly variability. As NCEP does not extend to earlier than 1948, to generate the 6-hourly variations in CRU-NCEP v4, the variability of a random year between 1948 and 1960 is applied to each year before 1948. This may partly explain why quality of the simulations before 1948 is not as good as afterwards.

If precipitation was higher than average during this El Niño event in the regions that usually experience drought instead (as indicated in the CRU/NCEP data), the reasons for these opposite anomalies could be understood. Li et al. (2013), using a 700-yr reconstruction of Nino3.4 have shown that the later decades of the 20th century were characterized by unusually high ENSO variability, while the 1940s registered a peak of low ENSO variance. During this period, Ashcroft et al. (2015) have found a break in the correlation of precipitation in south-eastern Australia and ENSO, associated with a positive phase

of the Inter-decadal Pacific Oscillation (Arblaster et al., 2002). The modulation of ENSO teleconnections in remote areas may imply a variable relationship between ENSO and the terrestrial sink that deserves deeper attention. Finally, it should also be noted that the 1940-42 very strong El Niño was followed by more than a decade with predominant La Niña conditions (Wolter and Timlin, 2011), coinciding with a negative phase of the Pacific Decadal Oscillation (Mantua and Hare, 2002), which may explain the persistence of an increased terrestrial-sink during the plateau period.

**4.2   The contribution of land-use change**

The different estimates of emissions from land-use change differ significantly during most of the 20th century, however their estimates diverge to a greater extent in the earlier decades of the 20th century. We find that the LUC emission estimates from the latest inter-model comparison exercise (TRENDY v4) presents good agreement with the bookkeeping data from Houghton (2003). The two BLUE datasets (Hansis et al., 2015) differ with the former two datasets by up to 1 Pg C yr$^{-1}$.

The discrepancies between BLUE and the other datasets likely result from the use of different methodologies, definitions and assumptions in each study (Gasser and Ciais, 2013; Hansis et al., 2015), such as, for example, the definitions of pasture areas or the way gross transitions are estimated. Moreover, the closer agreement of $E_{LUC-H}$ and $E_{LUC-DGVM}$ is incidental, as the models differ in the processes represented and definitions used (Houghton et al., 2012; Pongratz et al., 2014; Stocker and Joos, 2015). Such differences are considerable and their impact is of similar magnitude to, for instance, stopping deforestation or wood

harvest completely during 1940-1950, as estimated in idealized simulations using the OSCAR model. It is notable that the model estimates based on HYDE/Hurtt et al. (2011) show a stagnation of the previously rising land-use emission rates during the 1940s. The BLUE model shows that globally, emissions from cropland and pasture expansion slow down during the 1940s, while $CO_2$ uptake in abandoned land increases steeply. Net carbon sinks due to land-use change are thus created in parts of Europe, North America, and China, but they are not large enough to create an overall sink in the terrestrial biosphere (Fig. 3).




Land-use reconstructions rely on national inventories and agricultural statistics. While these sources of data are expected to have become reliable in recent decades, even then contradictory statistics are found at the country level and between reported and satellite-based estimates (Houghton, 2003). For the early 20th century statistics of deforestation, land-abandonment, and agricultural area are expected to be highly unreliable in many regions, due to the lack of inventories, e.g., in Amazonia (Imbach et al., 2015). Houghton (2003) has shown that revisions in recent inventories could account for regional differences in $E_{LUC}$ of about 0.3 Pg C yr$^{-1}$.

A major uncertainty results from all model studies applying land-use reconstructions that are based on FAO data for agricultural areas, which is available only from 1961 onwards. While Houghton (2003) included additional historical sources for some regions, the HYDE database (Klein Goldewijk et al., 2011) and thus the dataset by Hurtt et al. (2011) rely on extrapolating these country-level statistics back in time using population dynamics. In HYDE, the cropland and pasture values per capita are allowed to change 'slightly' prior to 1961 (Klein Goldewijk et al., 2011). Although changes in per-capita values between the 1940s and 1960s amount to only 1‰ when averaged over all countries, they may be as high as 50% in individual countries.

The uncertainty in the LUC emissions during the WW2 period thus remains high. Although statistics about food production, population or industrial output were kept because of their direct interest to the war effort management (Harrison, 2000), information about other processes relevant for $E_{LUC}$ may not be accurate (e.g., the impact of population mobilization for war and industry on land-abandonment, changes in wood harvest, etc). For example, the statistics for agricultural areas in the Soviet Union during 1940-1945 is almost absent. For the territory of the Russian Federation, reduction in the crop area during this period is estimated as 27% or about 25 Mha (Lyuri et al., 2010). The abandonment of cropland might be even higher for the most affected war territories of Ukraine and Belorussia, where agricultural production was severely reduced due to a shortage of manpower and destruction of infrastructure. The interruption in agricultural production extended beyond the war period, recovering only slowly. The crop area in Russia returned to the pre-war level only in the early 1950s (Lyuri et al., 2010). In China, the cropland area likely decreased during the war period, and only started to recover after 1949, according to Chinese Historical Cropland Database, which is not represented in HYDE dataset (He et al., 2013). A decade of reduced agricultural production and harvest in the war-stricken regions, not accounted in the HYDE dataset, could lead to substantial missing carbon uptake during this period.

The analysis of $\delta^{13}$ signatures corresponding to each of the idealized experiments show that differences in land-use change datasets as extreme as the ones tested here could still be compatible with the observed $\delta^{13}$ record. Changes in land use of the magnitude of our idealized tests are unlikely and effects of agricultural abandonment and halting of deforestation due to historical events have little effect on atmospheric $CO_2$ when persisting only for short periods of time (few decades or less), because model experiments suggest delayed emissions from past land-use change, in particular from soils, persist and regrowth takes time to reach its full potential (Pongratz et al., 2011). Brovkin et al. (2004) concluded that the stalling of atmospheric $CO_2$ during the 1940s was unlikely to have been caused by land-use changes. Still, the sensitivity experiments with OSCAR suggest that it is reasonable to expect that events not well represented (or included at all) in the current LUC reconstructions may provide a non-negligible fraction of the 0.4-1.5 Pg C yr$^{-1}$ required for reconstructions to match the $CO_2$ record during the period.





### 4.3 Other sources of uncertainty

Another process that could potentially contribute to a further increase in the terrestrial sink is the impact of nitrogen deposition in net primary productivity. Thomas et al. (2010) have shown that nitrogen deposition stimulated carbon sequestration in temperate forests in the USA during the 1980s and 1990s, with stronger sensitivity of carbon accumulation to lower lev-
els of nitrogen inputs. The authors estimated that nitrogen deposition could increase carbon storage in ecosystems by ca. 0.3 Pg C yr$^{-1}$.

The increase in fossil fuel burning due to industrial expansion and the beginning of the automobile era produced strong changes in nitrogen deposition. The strong initial response of plants to high levels of nutrients could have produced a sudden increase in the terrestrial sink, followed by saturation due to soil acidification as deposition rates persisted (Gundersen et al.,
2006) and other limitations such as phophorus came into play (Vitousek et al., 2010). At present, DGVMs still struggle to represent realistically the interactions between ecosystems, and the nitrogen and phosphorus cycles. Nevertheless, the DGVMs used here that include the nitrogen cycle (CLM4.5, OCN and VISIT) estimate very similar values for the terrestrial sink during the plateau period, and slightly stronger than the inter-model mean (Table 2).

### 5 Conclusions

This work has used the currently available estimates of sources and sinks of $CO_2$ during the 20th century and their associated uncertainties to gain insight into the temporary stabilization of atmospheric $CO_2$ concentration observed during the 1940s until mid-1950s, as well as evaluating the mechanism previously identified as the main driver of such stabilization.

Our results show that, although the oceans are likely to have contributed, they cannot by themselves provide the complete explanation of the 1940s plateau. A strong terrestrial sink is also required to match the observed stalling in atmospheric $CO_2$
during the period. Further work is required to narrow the uncertainty in the carbon budget components in order to identify other processes that might help to explain the 1940s plateau.

However, the discrepancies between observations and the carbon budget estimated using independent reconstructions of each component are not particular to the 1940s. This indicates that efforts to narrow down the uncertainty of each term of the carbon budget are required.

The relationship between reconstructed terrestrial and ocean fluxes with the climate anomalies observed during the early 20th century deserve greater attention. Given the large difference between estimates of ocean flux anomalies in response to climate variability, a new initiative is needed to better characterize $CO_2$ fluxes in the ocean during the 20th century, e.g., by forcing the ocean circulation models with climate reconstructions. In the case of the terrestrial sink, other processes currently not included in the models or in the LUC reconstructions may have contributed to the plateau. The effects of fire occurrence,
changes in nutrient availability and the devastating socioeconomic consequences of WW2 are examples of processes currently not well represented in the models.

It should be noted that the high-resolution Law Dome record is unique in its precision and quality. However, the large measurements errors in even the best $\delta^{13}$C ice core data currently available make it difficult to accurately quantify variations in





the oceanic and terrestrial sinks. In high accumulation sites such as DE08, new measurements of $\delta^{13}$ with improved accuracy should reveal the high-resolution information contained in the ice sheet and reduce the scatter of current estimates. It would also be advantageous to get another insight into atmospheric $CO_2$ and $\delta^{13}C$ changes during 1940s from a second high-resolution core.

5   This study thus allows us to identify a number of key aspects of the global carbon budget that require deeper attention, if we are to better characterize the coupled carbon-climate variability in the 20th century.

*Author contributions.* A.B. conducted the analysis and wrote the manuscript. P.C., J.B., N.V, L.B., S.P. and V.B. helped in conceiving the analyses and provided expert advice. T.G. is responsible for the OSCAR model and helped to perform the analysis. J.P. is responsible for the BLUE data and provided expert advice. C.M.T. helped analysing the ice-core data. All authors contributed to the revision of the paper.

10   *Acknowledgements.* The authors would like to thank S. Khatiwala for providing the dataset of anthropogenic $CO_2$ ocean uptake. Double-deconvolution data from Joos et al. (1999) is publicly available at: http://www.climate.unibe.ch/~joos/OUTGOING/JOOSetal_GRL_1999/. C.M.T. acknowledges the support of the Australian Climate Change Science Program. The work is supported by the Commissariat à l'énergie atomique et aux énergies alternatives (CEA), France.





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





**Table 1.** Summary of the dyamics global vegetation models used to estimate $L_{DGVM}$. More details about the way each model represents LUC may be found in Le Quéré et al. (2015).

| Model | Spatial resolution | Vegetation | Fire | N-cycle | Reference |
|---|---|---|---|---|---|
| CLM4.5 | 1°×1° | Imposed | Y | Y | Oleson et al. (2013) |
| JULES | 1.88°×1.25° | Dynamic | N | N | Cox (2001) |
| JSBACH | 0.5°×0.5° | Imposed | N | N | Reick et al. (2013) |
| LPJmL | 0.5°×0.5° | Dynamic | Y | N | Bondeau et al. (2007) |
| LPJ-GUESS | 0.5°×0.5° | Dynamic | Y | N | Sitch et al. (2003) |
| OCN | 1°×1° | Imposed | Y | Y | Zaehle et al. (2010) |
| ORCHIDEE | 2°×2° | Imposed | Y | N | Krinner et al. (2005) |
| VISIT | 0.5°×0.5° | Imposed | N | Y | Ito (2010) |

**Table 2.** Terrestrial sink during the plateau period (1940-1950) and during the El Niño event of 1940-1942, estimated by the set of DGVMs. Positive values of L correspond to increased $CO_2$ uptake by terrestrial ecosystems and negative values to a terrestrial $CO_2$ source. Values in Pg C year$^{-1}$.

| Model | L (1940-1950) | L (1940-1942) |
|---|---|---|
| CLM4.5 | 0.79 | 0.81 |
| JULES | 0.72 | -0.29 |
| JSBACH | 1.14 | 1.27 |
| LPJ-GUESS | 0.50 | 0.49 |
| LPJmL | -0.43 | 0.09 |
| OCN | 0.89 | 0.30 |
| ORCHIDEE | 1.2 | 0.80 |
| VISIT | 0.85 | 0.57 |

**Table 3.** Difference between reconstructed and observed AGR ($\Delta$AGR, in Pg C yr$^{-1}$) during the periods 1940-1950 (positive values indicate an over-estimation by the reconstructions). $\Delta$AGR$_J$ corresponds to the reconstruction using $E_{FF}$, $O_J$ and $B_J$ as in Figure 2 and the other $\Delta AGR$ values to the reconstructions based on $E_{FF}$, $O_K$, the inter-model median value of the land-sink estimated by DGVMs (S2) and the different estimations of $E_{LUC}$, as in Figure 3.

| Set | $\Delta AGR$ |
|---|---|
| $\Delta AGR_J$ | 0.1±0.7 |
| $\Delta AGR_H$ | 0.9±0.8 |
| $\Delta AGR_{DGVM}$ | 1.2±1.0 |
| $\Delta AGR_B$ | 2.0±0.8 |
| $\Delta AGR_{Blc}$ | 1.5± 0.8 |





**Table 4.** Constants and parameters used to calculate resulting $\delta 13C$ from the OSCAR simulations.

|  | Description | Value | Reference |
|---|---|---|---|
| $NPP_0$, $LB_0$ | gross terrestrial fluxes in 1700 | $54 PgC.year^{-1}$ | Running (2012) |
| $OA_0$, $AO_0$ | gross oceanic fluxes in 1700 | $73 PgC.year^{-1}$ | Naegler et al. (2006) |
| $\delta^f$ | $\delta^{13}C$ of fossil fuel $CO_2$ | -24 (1750) to -28 (2010) | Andres et al. (1994) |
| $\delta^o$ | $\delta^{13}C$ of ocean surface water | 2.5 (1750) to 1.5 (2010) | Hellevang and Aagaard (2015) |
| $\delta^{lb}$ | $\delta^{13}C$ of the terrestrial biosphere | -25 | Hellevang and Aagaard (2015) |
| $\epsilon^{lb}$ | isotopic fractionation of tthe terrestrial biosphere | -7 | Ciais et al. (2014) |
| $\epsilon^o$ | isotopic fractionation between the air and ocean | 0 | Hellevang and Aagaard (2015) |

**Table 5.** Average global LUC transitions (in Mha $yr^{-1}$) during 1940-1950 from Hurtt et al. (2011), used in the OSCAR model default setup.

|  | Transition to | | | | |
|---|---|---|---|---|---|
|  | Des.+ Urb. | For. | Grass. + Shrub. | Crop. | Past. |
| Des.+ Urb. | - | - | - | 1.6 | 9.5 |
| For. | - | - | - | 2.1 | 4.9 |
| Grass. +Shrub. | - | - | - | 4.4 | 15.0 |
| Crop. | 0.2 | 0.7 | 0.8 | - | 2.4 |
| Past. | 2.6 | 6.2 | 4.6 | 1.5 | - |

**Table 6.** Maximum decadal anomalies of ocean $CO_2$ uptake in the IPSL-CMA5 simulations (PgC.year$^{-1}$ per decade) and corresponding anomaly in tropical SST (°C) in the Nino3.4 region. The annual values of the ocean fluxes are filtered using the same smoothing as the one applied to AGR, based on the air-age distribution filter from Trudinger et al. (2003). The SST anomaly is calculated as the average departure of the filtered SST data from a 30-yr long reference period.

| Realisation | Time | $O_{anom}$ | $SST_{anom}$ |
|---|---|---|---|
| IPSL-CMA5-LR r1i1p1 | 1932 | 0.08 | 0.01 |
| IPSL-CMA5-LR r2i1p1 | 1885 | 0.06 | -0.06 |
| IPSL-CMA5-LR r3i1p1 | 1889 | 0.11 | -0.08 |
| IPSL-CMA5-LR r4i1p1 | 1930 | 0.13 | 0.01 |
| IPSL-CMA5-LR r5i1p1 | 1991 | 0.14 | -0.06 |
| IPSL-CMA5-LR r6i1p1 | 1895 | 0.12 | 0.02 |
| IPSL-CMA5-MR r1i1p1 | 1991 | 0.22 | -0.05 |
| IPSL-CMA5-MR r2i1p1 | 1918 | 0.20 | -0.06 |
| IPSL-CMA5-MR r3i1p1 | 1976 | 0.17 | -0.02 |





**Table 7.** Average difference in $E_{LUC}$ and atmospheric $CO_2$ growth rate between the OSCAR standard run and the simulations using different LUC hypothetical scenarios during 1940-1950, in Pg C yr$^{-1}$.

| Test (from 1940 until 1950) | $\Delta E_{LUC}$ | $\Delta$ AGR |
|---|---|---|
| T1 - stop deforestation | 0.51 | 0.39 |
| T2 - double forest expansion | 0.07 | 0.06 |
| T3 - stop crop+pasture expansion | 0.49 | 0.37 |
| T4 - stop wood harvest | 0.23 | 0.17 |

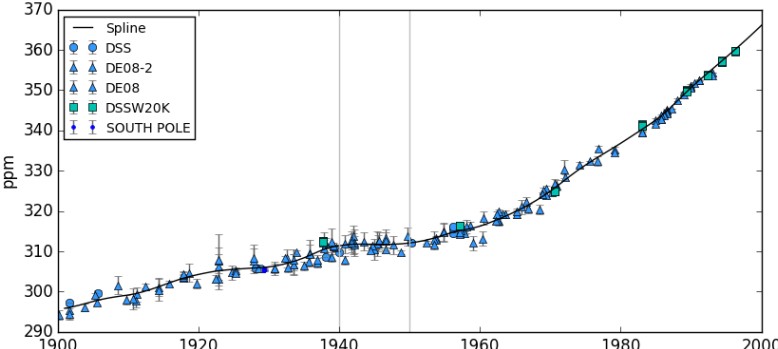

**Figure 1.** Atmospheric $CO_2$ concentration in the Law Dome ice core and firn record from Rubino et al. (2013) and respective uncertainties (markers and whiskers), and the spline-fit applied to the data following Enting et al. (2006), which attenuates by 50% variations of c.a. 23 years. The period corresponding to the plateau is highlighted between vertical grey lines.



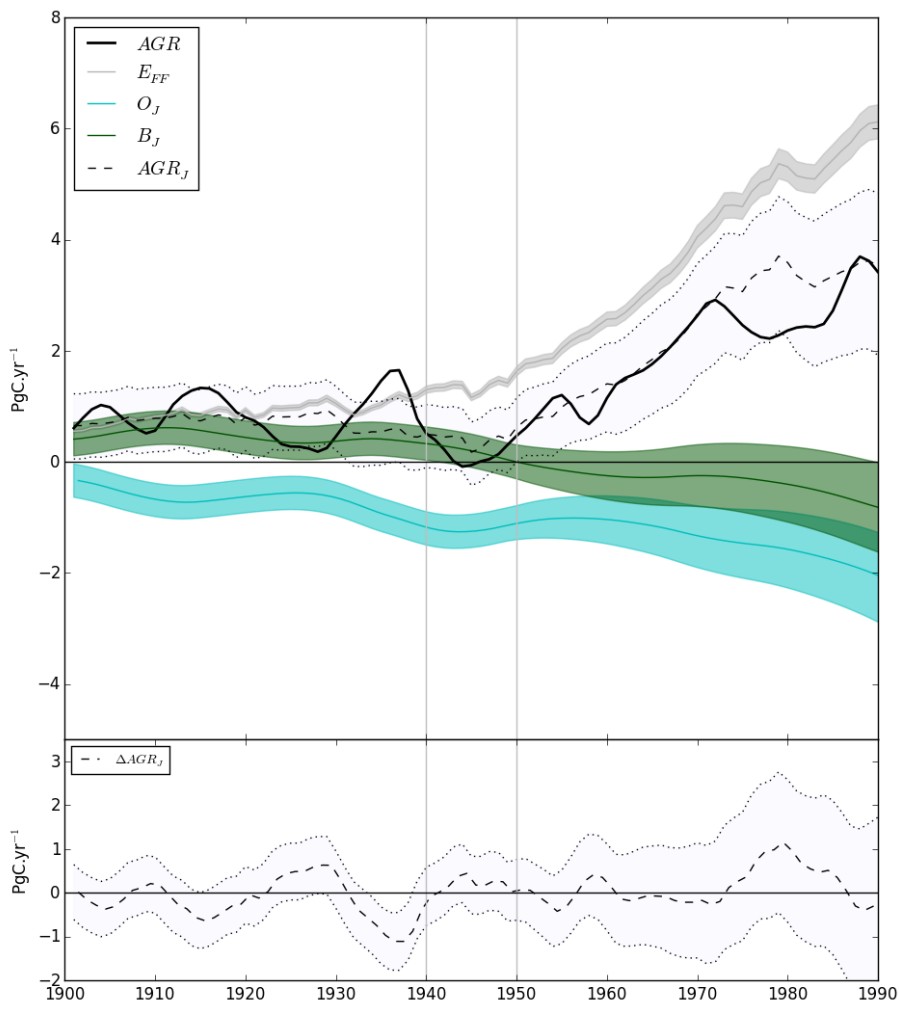

**Figure 2.** Atmopheric $CO_2$ growth rate (AGR) from the observational record, calculated from the spline fit in Figure 1 (black line, top panel). Fossil fuel emissions from the CDIAC database and respective uncertainty ($E_{FF}$), and the reconstruction of ocean and biospheric fluxes from Joos et al. (1999), $O_J$ and $B_J$ respectively (filled areas in top panel). The resulting balance from the latter three datasets ($AGR_J$) and uncertainty is shown in the top panel (dashed and dotted lines, respectively), and the corresponding difference between AGR and $AGR_J$ is shown in the bottom panel.





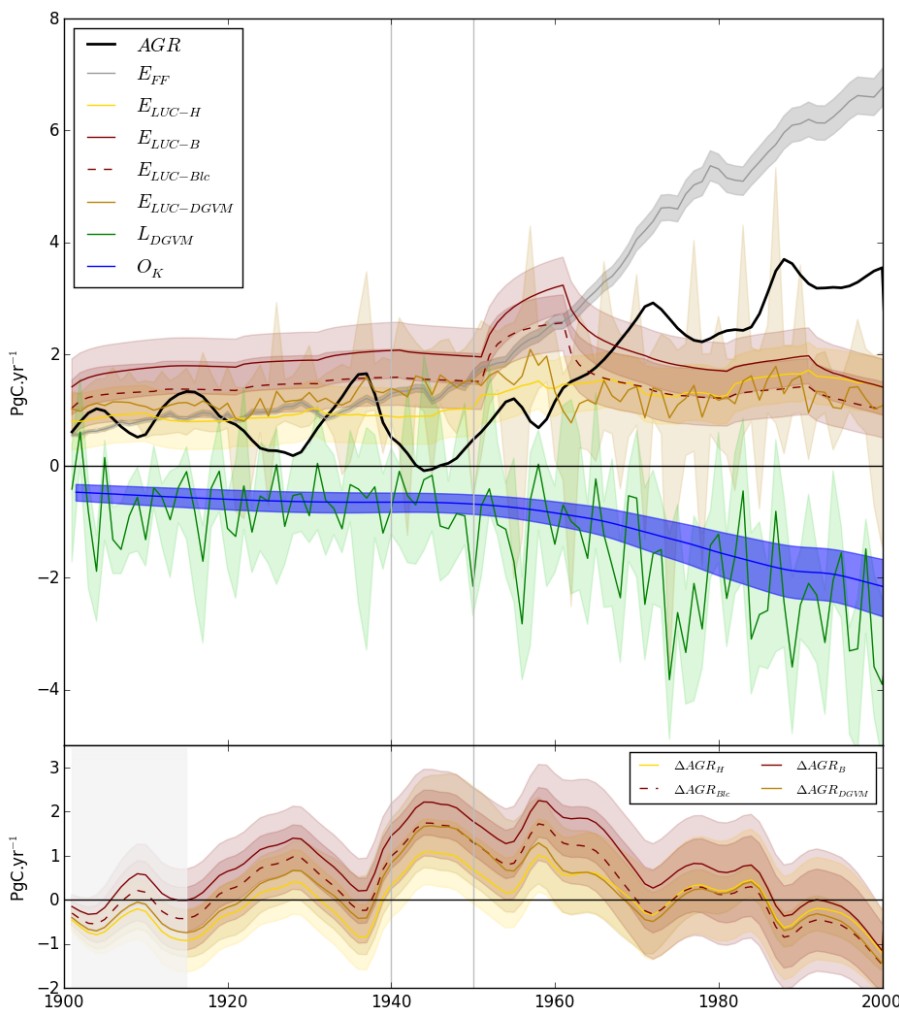

**Figure 3.** As in Figure 2 but for the independent estimates of sources and sinks: $E_{FF}$ from CDIAC, $E_{LUC}$ from Houghton (H), BLUE (B) and BLUE with lower C-stock changes (Blc) and DGVMs forced with LUC, ocean from Khatiwala et al. (2009) reconstruction and land-sink as estimated by DGVMs forced only by $CO_2$ and climate. In the bottom panel, the difference between observed AGR and $AGR_H$, $AGR_B$, $AGR_{Blc}$ and $AGR_{DGVM}$ is shown.





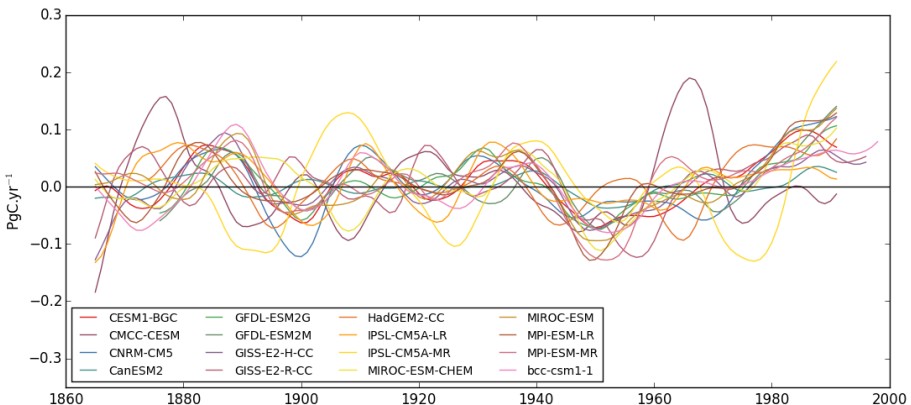

**Figure 4.** Variability in the global $CO_2$ uptake by the oceans, estimated by the group of CMIP5 climate models for the historical simulation, with prescribed atmospheric $CO_2$, as well as solar radiation variability, sulphate aerosols and volcanic eruptions. The annual values of the ocean fluxes are filtered using the same smoothing as the one applied to AGR, based on the air-age distribution for $CO_2$ at DE08 from Trudinger et al. (2003).

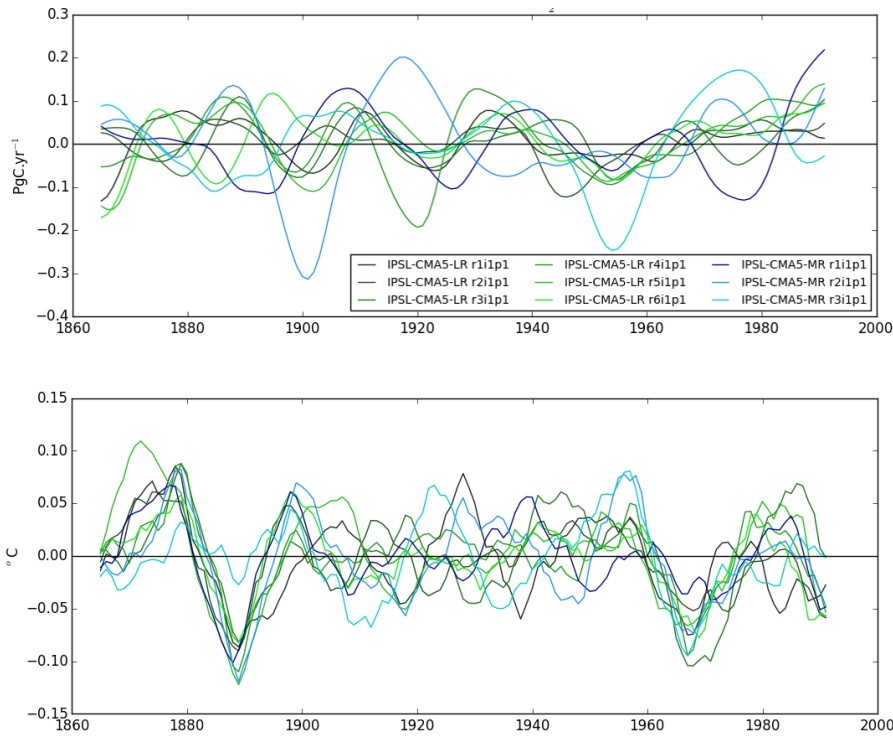

**Figure 5.** As in Figure 4, but for six different realisations from IPSL-CM5A-LR and three from IPSL-CM5A-MR (top panel) and the corresponding SST temperature anomalies in the Nino3.4 region. The SST anomaly is calculated as the 10-yr moving average departure of the SST data from a 30-yr long reference period.





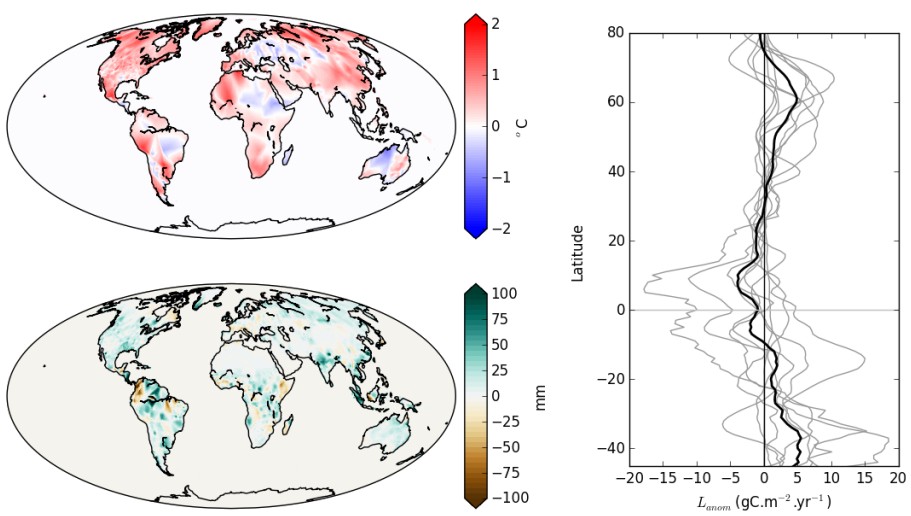

**Figure 6.** Response of the terrestrial ecosystems to the climate anomalies during the plateau period, simulated by the DGVMs. Temperature (left top) and precipitation (left bottom) anomaly fields during 1940-50 (relative to 1900-1930), and the corresponding latitudinal anomaly of $L_{DGVM}$ estimated by each model (grey lines) and the multi-model average (right panel).





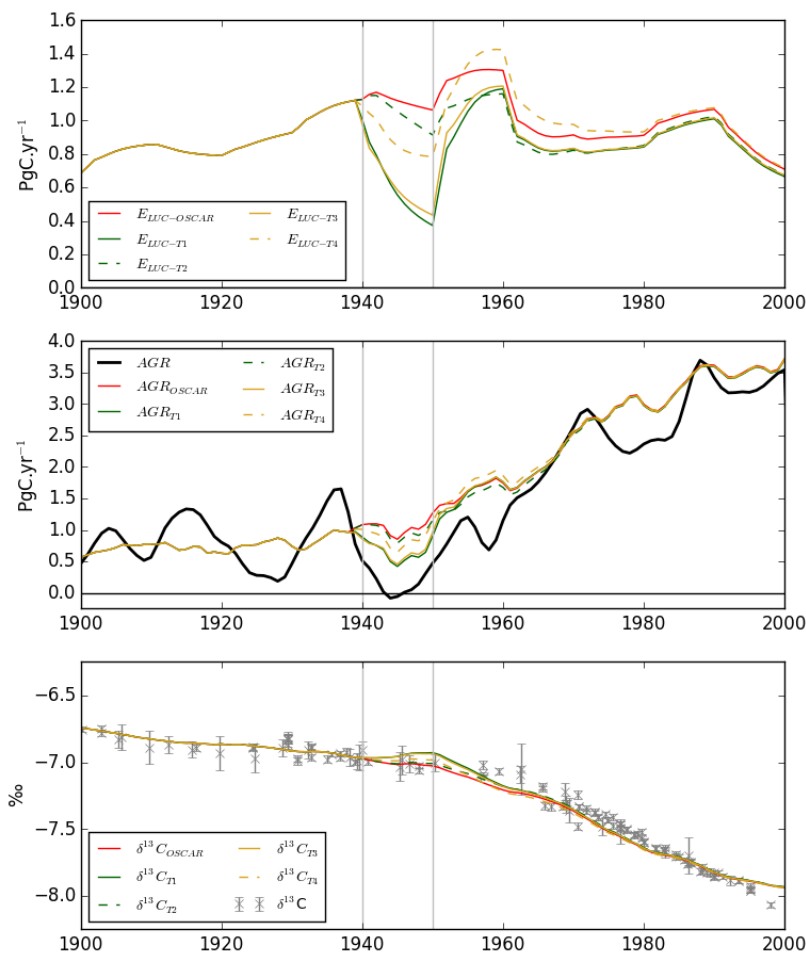

**Figure 7.** Resulting $E_{LUC}$ from OSCAR simulations for hypothetical scenarios about changes in LUC during 1940-1950 (top). In T1 forest conversion is set to zero (green solid), in T2 the rate of forest expansion during the period is doubled (green dashed), in T3 cropland and pasture expansion are stopped (yellow solid) and in T4 wood harvest is set to zero (yellow dashed). The $E_{LUC}$ from each test are compared with the LUC emissions in the standard OSCAR simulation (red). The atmospheric $CO_2$ growth-rate (AGR) resulting from standard OSCAR and each test are compared with the ice-core record (center). The $\delta^{13}C$ values corresponding to each test (bottom) are compared with $\delta^{13}C$ from the ice-core record and the corresponding uncertainty (markers and errorbars).