# Peer review of "Re-evaluating the 1940s CO2 plateau"

_Biogeosciences, 2016_

## Referee Comment (RC1) · R.A. Houghton (Referee) · 14 Jun 2016

This paper represents a comprehensive analysis of the processes that may have been responsible for the plateau in CO2 growth rates that occurred during the 1940s and early 1950s. The mismatch between model reconstructions and observations during this period was 0.9 to 2.0 PgC yr-1. Ocean carbon models from CMIP5 suggest that natural variability in the oceans could have accounted for no more than 0.5 PgC yr-1, while TRENDY models suggest that the land's response to CO2 and a strong El Nino would not have accounted for the necessary carbon sinks on land. Using the OSCAR model, the authors explored whether changes in land use (LUC) might have led to large terrestrial sinks. They found that LUC might have provided the necessary land sinks for carbon, likely through the effects of socio-economic changes during WWII, but that such conditions are not well captured by existing LUC analyses. While I completely agree with the statement that many activities associated with wars and

economic disasters are not well captured by land-use statistics, I am surprised that stopping deforestation and logging could have an effect large enough to create sinks of 1-2 PgC yr-1. Sources that large are believable, but sinks that large would require large areas of regrowth, largely because per hectare sinks from forest growth are generally slow in comparison with per hectare sources from harvest and deforestation. Would the changes during WWII have been widely enough distributed to affect Europe, the USSR, China as well as SE Asia and perhaps other regions? Regardless, the authors are to be commended for the multiple and penetrating analyses carried out for this exploration. They have a solid understanding of land-use data sets. While the observed plateau in CO2 growth rates during the 1940s may appear small in the scheme of things, it is not so small as to be easily explained. This analysis is interesting, packed with information from many disciplines, and impressive.

———————————————————

---

## Referee Comment (RC2) · Anonymous Referee #2 · 21 Jul 2016

The study analyses the causes of the 1940s atmospheric CO2 flattening measured in ice core bubbles from Law Dome (East Antarctica). The CO2 plateau during the 1940-1950 decade is one of the significant (and still unexplained) features of the carbon cycle over the last centuries and millennia. Coupled Climate Carbon Cycles Models would benefit from an understanding of the causes of the 1940s CO2 flattening, as they are likely to improve their accuracy in estimating future climate-carbon cycle changes. The subject of the paper is thus very relevant for biogeochemical investigations and fits within the scope of the journal. There are two novel aspects: 1) combining informations from a number of different studies (investigating fossil fuel and land use change CO2 emissions, as well as ocean and land CO2 sequestration) in a comprehensive, review-like type of study of the causes of the 1940s CO2 plateau; 2) using the OSCAR model to explore whether land use changes could have led to a significant land sink. Even though the reasons of the 1940s CO2 plateau remain elusive, the conclusions reached are significant as the authors explain that the ocean

sink alone cannot provide the complete explanation, and a strong contribution from the land sink is needed. I found the approach used by Bastos et al. comprehensive and clear. All calculations use state-of-the-art models and valid assumptions. The results are supportive of the coclusions. Nevertheless, I would have liked the authors to be more critical with the estimate of fossil fuel emissions, which they assume are accurate within the given uncertainties. Is it possible that the estimates provided by the CDIAC are biased? The description of calculations are complete and precise to allow their reproduction. The language and the presentation are clear. However, I have reported several specific comments to improve the paper in the attachment.

Please also note the supplement to this comment:
http://www.biogeosciences-discuss.net/bg-2016-171/bg-2016-171-RC2-supplement.pdf

**Supplement:**

[revised manuscript text omitted]

---

## Referee Comment (RC3) · I G Enting (Referee) · 1 Aug 2016

The anomalous 1940's dip in CO2 was noted twenty years ago and a definitive explanation is still lacking. By presenting an extensive range of comparisons, this paper brings into focus the difficulties. As such, it is a valuable contribution and, subject to clarifying the issues noted below, is suitable for publication in Biogeosciences.

I would have liked a clearer statement, for each of the comparisons, of the "boundary conditions" applicable for each case, i.e. what is being assumed as "fixed" in each case (e.g. single deconvolutions assume fixed (i.e. time-invariant) ocean response, while double deconvolutions assume invariant mixed-layer response).

With regard to the results presented in fig 2, the authors note that they are comparing (by taking the difference) AGR based on a 25-yr cutoff spline and AGR based on splines with a longer cutoff. This can be simply described as applying a band-pass filter to

AGR. Saying this explicitly might help the reader, but it also suggests that the analysis in figure 2 adds little to the overall analysis in the paper.

A minor point is the implication that zero AGR requires zero fossil emissions (or a change on uptake processes). This is not correct. Zero growth rate can be achieved by a rapid reduction in emissions, with uptake processes responding to higher atmospheric concentrations. Note for example stabilisation calculations, or the discussion by Gloor 2010 (Atmos Phys Chem, 10, 7739). (This a case of poor wording and it in no way invalidates the overall analysis in the paper).

---

## Author Comment (AC2) · 5 Aug 2016

We would like to thank the referee for the careful review and detailed comments, that help improving the quality of the manuscript.

We acknowledge that it is possible that fossil fuel emissions estimates have higher uncertainty during the WW2 period. An alternative estimate of E$_{FF}$ for the 20th century may be found in Mohr et al. (2015). Their estimates for the 1940-1950 differ by about 0.1PgC/yr from the CDIAC ones, i.e. 7.5%, slightly more than the 5% uncertainty range defined by the CDIAC. Quilcaille et al. (2016, conference proceedings) calculate that uncertainty in datasets and the different methodologies to estimate E$_{FF}$ from statistics of fossil fuel extracted may increase total E$_{FF}$ uncertainty up to 11%. Even considering an uncertainty range as high as the one suggested by Quilcaille et al. (2016), the difference would be 0.15PgC/yr, which would not suffice to explain the CO$_2$ stabilisation in the 1940s. The authors, nevertheless, agree that it is worth including a note about

the subject in the revised version of the manuscript.

The referee's proposed corrections will also be included in the revision.

**References**

Mohr, S. H., et al. Projection of world fossil fuels by country. Fuel 141 (2015): 120-135.

Quilcaille, Yann, et al. Uncertainty in projected climate change caused by methodological discrepancy in estimating $CO_2$ emissions from fossil fuel combustion. EGU General Assembly Conference Abstracts. Vol. 18. 2016.

---

## Author Comment (AC3) · 5 Aug 2016

**The anomalous 1940's dip in CO$_2$ was noted twenty years ago and a definitive explanation is still lacking. By presenting an extensive range of comparisons, this paper brings into focus the difficulties. As such, it is a valuable contribution and, subject to clarifying the issues noted below, is suitable for publication in Biogeosciences.**

The authors would like to thank the referee for the review and for highlighting important aspects that needed improvement.

**RC1: I would have liked a clearer statement, for each of the comparisons, of the "boundary conditions" applicable for each case, i.e. what is being assumed as "fixed" in each case (e.g. single deconvolutions assume fixed (i.e. time-invariant) ocean response, while double deconvolutions assume invariant mixed-layer re-**

**sponse).**

AR: We agree with the referee and this point will be included in the revised version of the manuscript.

**RC2: With regard to the results presented in fig 2, the authors note that they are comparing (by taking the difference) AGR based on a 25-yr cutoff spline and AGR based on splines with a longer cutoff. This can be simply described as applying a band-pass filter to AGR. Saying this explicitly might help the reader, but it also suggests that the analysis in figure 2 adds little to the overall analysis in the paper.**

AR: The purpose of Fig. 2 is two-fold:

i) To present the estimates of the ocean sink from the double-deconvolution by Joos et al. (1999), which are then used as a reference for the possible contribution of the ocean to the increased $CO_2$ uptake during the 1940s;

ii) To show that discrepancies between AGR reconstructed from the different terms and the observations are still to be expected, partly because of the different choices of smoothing, but also for the reasons discussed in Section 2.3.1.

Therefore, we consider that Fig. 2 is worth keeping, although we agree that a sentence clarifying the problem may be included in the revision.

**RC3: A minor point is the implication that zero AGR requires zero fossil emissions (or a change on uptake processes). This is not correct. Zero growth rate can be achieved by a rapid reduction in emissions, with uptake processes responding to higher atmospheric concentrations. Note for example stabilisation calculations, or the discussion by Gloor 2010 (Atmos Phys Chem, 10, 7739). (This a case of poor wording and it in no way invalidates the overall analysis**

**in the paper)**

AR: We thank the referee for pointing this important issue, as the phrasing was indeed not the most correct. However, given the datasets available, it is not likely that an abrupt decrease in E$_{FF}$ as large as the one needed to stabilise atmospheric $CO_2$ might have occurred during the 1940s (see AR to Referee 2). Nevertheless, we agree that the sentence can be reformulated in the revised version of the manuscript, together with a reference to Gloor et al. (2010).

---

## Author Response (AR1)

**Laboratoire des Sciences du Climat et de l'Environnement**
LSCE (UMR 8212)

Dear Editor,

We are now submitting the revised version of the manuscript "Re-evaluating the 1940s CO2 plateau" submitted to Biogeosciences (bg-2016-171). We have incorporated all the corrections suggested by Reviwer #2 and addressed the questions raised by the three referees.

Please find below the responses to the referees, now including the corresponding changes made to the original manuscript. We also include a version of the document with track-changes, to account for all the small changes made to the original text.

On behalf of the authors.

Yours sincerely,
Ana Bastos*

* Corresponding author
ana.bastos@lsce.ipsl.fr
IPSL – LSCE (CEA CNRS UVSQ)
Centre d'Etudes Orme des Merisiers
91191 Gif sur Yvette France
01.69.08.34.03

**Re-evaluating the 1940s plateau**
Author's response

Referee #1 (Richard Houghton)

**RC: This paper represents a comprehensive analysis of the processes that may have been responsible for the plateau in CO2 growth rates that occurred during the 1940s and early 1950s. The mismatch between model reconstructions and observations during this period was 0.9 to 2.0 PgC yr-1. Ocean carbon models from CMIP5 suggest that natural variability in the oceans could have accounted for no more than 0.5 PgC yr-1, while TRENDY models suggest that the land's response to CO2 and a strong El Nino would not have accounted for the necessary carbon sinks on land. Using the OSCAR model, the authors explored whether changes in land use (LUC) might have led to large terrestrial sinks. They found that LUC might have provided the necessary land sinks for carbon, likely through the effects of socio-economic changes during WWII, but that such conditions are not well captured by existing LUC analyses. While I completely agree with the statement that many activities associated with wars and economic disasters are not well captured by land-use statistics, I am surprised that stopping deforestation and logging could have an effect large enough to create sinks of 1-2 PgC yr-1. Sources that large are believable, but sinks that large would require large areas of regrowth, largely because per hectare sinks from forest growth are generally slow in comparison with per hectare sources from harvest and deforestation. Would the changes during WWII have been widely enough distributed to affect Europe, the USSR, China as well as SE Asia and perhaps other regions? Regardless, the authors are to be commended for the multiple and penetrating analyses carried out for this exploration. They have a solid understanding of land-use data sets. While the observed plateau in CO2 growth rates during the 1940s may appear small in the scheme of things, it is not so small as to be easily explained. This analysis is interesting, packed with information from many disciplines, and impressive.**

AR: The authors would like to thank the referee for the encouraging comments. In Section 3.2.3, the use of different extreme LUC scenarios for the plateau period in OSCAR is intended to provide an estimate of how much $E_{LUC}$ may theoretically contribute to the required sink, given the $\delta^{13}C$ record constraint. We show that given its high uncertainty, even extreme changes as the idealised experiments defined could be compatible with the $\delta^{13}C$ record. Nevertheless, the authors would like to point out that the effect of war-related mortality and migrations is, very likely, not negligible. For example, Vuichard et al. (2008) have shown that land-abandonment after the collapse of the former Soviet Union, estimated to be of about 20 million hectares led to a small, but still significant, sink of about 64TgC in 10 years.

In Hurtt et al. (2011), a 6 million ha decrease of crop area between 1940 and 1950 in the Soviet Union (figure below) is reported, which appears to be rather small, considering the war-time demography and economics.

[Figure]

Fig. 1 Changes in crop area in the Former Soviet Union during the 20[th] century reported by Hurtt et al. (2011).

Analysis of Lyuri et al. (2010) based on agricultural data archives revealed much bigger drop in ca. 30 Mha during 1940-1950 just in Russia (figure below), not accounting for a decrease in crop areas in Ukraine and Belorussia severely affected by the war.

[Figure]

*Fig. 2 Changes in crop (sown) area in Russia with two sharp drops during periods of the civil war and the WWII (solid line). Dashed line: hypothetical scenario of crop area development in the absence of these two crises. Reproduced from Lyuri et al. (2010, Fig. 2.28).*

The number of war-related deaths is estimated to be 26.6 million people, about 14% of the population (Harrison, 2000) and, with the re-location of the industry from the western front to the eastern provinces, about 10 million people are estimated to have been evacuated from the western areas (Nove, 1989). Furthermore, agricultural output is estimated to have fallen by up to 60% during the peak of the war (Nove, 1989). Also, at the time of WW2, the reliance of Russian population on fuel wood was likely much larger than in the last decades of the Soviet Union. The huge decrease in population also decreased harvest pressure on forests and woodlands, so our "LUC hypothesis" is in reality a land management hypothesis.

Likewise, the war-related mortality during WWII in China is estimated to be of about 14million people (Mitter, 2013), although its impacts on agriculture and economy are not so well known.

Although these changes were rather fast and were followed by recovery in after the war, their effects in carbon stocks and on gross LUC emissions might not be negligible. As a thought experiment, and considering the estimates by deB Richter Jr & Houghton (2011) of gross LUC fluxes for the first years of the 21st century, a suppression of present gross LUC emissions would provide an additional sink of 2.5PgC/yr.

The authors acknowledge that detailed information regarding land-use are hard to obtain for the earlier 20th century, and especially during the war period and therefore this exercise might remain speculative. Nevertheless, further efforts to obtain detailed cropland area in key countries such as the ones severely affected by war could potentially shed some light on the impact of fast but devastating events on the carbon dynamics of ecosystems.

We have addressed these points in the discussion section (page 18, line 28 to page 19, line 8), which now reads:

*"For example, the statistics for agricultural areas in the Soviet Union during 1940-1945 is almost absent. Hurtt et al. (2011) report a 6.6 million ha decrease of crop area between 1940 and 1950 in the former Soviet Union, but (Lyuri et al., 2010) estimated a reduction in crop area in the territory of the Russian Federation of 27% or about 25 Mha, for the same period. The number of war-related deaths is*

*estimated to be 26.6 million people, about 14% of the population (Harrison, 2000) and agricultural output is estimated to have fallen by up to 60% during the peak of the war (Nove, 1982). Furthermore, with the re-location of the industry from the western front to the eastern provinces, about 10 million people are estimated to have been evacuated from the western areas (Nove, 1982). Thus, the abandonment of cropland might be even higher for the most affected war territories of Ukraine and Belorussia, where agricultural production was severely reduced due to a shortage of manpower and destruction of infrastructure. The interruption in agricultural production extended beyond the war period, recovering only slowly. The crop area in Russia returned to the pre-war level only in the early 1950s (Lyuri et al., 2010). Also, at the time of WW2, the reliance of Russian population on fuel wood was likely much larger than in the last decades of the Soviet Union.*

*In China, the war-related mortality during WW2 in China is estimated to be of about 14 million people (Mitter, 2013) and mass migration movements were also reported (Lary, 2010). The cropland area likely decreased during the war period, and only started to recover after 1949, according to Chinese Historical Cropland Database, which is not represented in HYDE 3.1 dataset (He et al., 2013). A decade of reduced agricultural production and harvest in the war-stricken regions, not accounted in the HYDE 3.1 dataset, could lead to substantial missing carbon uptake during this period."*

deB Richter Jr, Daniel, and R. A. Houghton. "Gross CO2 fluxes from land-use change: implications for reducing global emissions and increasing sinks."*Carbon Management* 2.1 (2011): 41-47.

Mitter, Rana. *Forgotten Ally: China's World War II, 1937-1945*. Houghton Mifflin Harcourt, 2013.

Lyuri D.I., Goryachkin S.V., Karavaeva N.A., Denisenko E.A., Nefedova T. G., 2010. *Dynamics of Agricultural lands of Russia in XX century and Postagrogenic Restoration of vegetation and soils.* Moscow, GEOS, 416 p. (In Russian)

Nove, Alec. *An economic history of the USSR*. IICA, 1982.

Vuichard, Nicolas, et al. "Carbon sequestration due to the abandonment of agriculture in the former USSR since 1990." *Global Biogeochemical Cycles*22.4 (2008).

Referee #2

**RC: The study analyses the causes of the 1940s atmospheric CO2 flattening measured in ice core bubbles from Law Dome (East Antarctica). The CO2 plateau during the 1940-1950 decade is one of the significant (and still unexplained) features of the carbon cycle over the last centuries and millennia. Coupled Climate Carbon Cycles Models would benefit from an understanding of the causes of the 1940s CO2 flattening, as they are likely to improve their accuracy in estimating future climate-carbon cycle changes. The subject of the paper is thus very relevant for biogeochemical investigations and fits within the scope of the journal. There are two novel aspects: 1) combining informations from a number of different studies (investigating fossil fuel and land use change CO2 emissions, as well as ocean and land CO2 sequestration) in a comprehensive, review-like type of study of the causes of the 1940s CO2 plateau; 2) using the OSCAR model to explore whether land use changes could have led to a significant land sink. Even though the reasons of the 1940s CO2 plateau remain elusive, the conclusions reached are significant as the authors explain that the ocean sink alone cannot provide the complete explanation, and a strong contribution from the land sink is needed. I found the approach used by Bastos et al. comprehensive and clear. All calculations use state-of-the-art models and valid assumptions. The results are supportive of the coclusions. Nevertheless, I would have liked the authors to be more critical with the estimate of fossil fuel emissions, which they assume are accurate within the given uncertainties. Is it possible that the estimates provided by the CDIAC are biased? The description of calculations are complete and precise to allow their reproduction. The language and the presentation are clear.**

AR: We would like to thank the referee for the careful review and detailed comments, that help improving the quality of the manuscript.

We acknowledge that it is possible that fossil fuel emissions estimates have higher uncertainty during the WW2 period. An alternative estimate of $E_{FF}$ for the 20[th] century may be found in Mohr et al. (2015). Their estimates for the 1940-1950 differ by about 0.1PgC/yr from the CDIAC ones, i.e. 7.5%, slightly more than the 5% uncertainty range defined by the CDIAC. Quilcaille et al. (2016, conference proceedings) calculate that uncertainty in datasets and the different methodologies to estimate $E_{FF}$ from statistics of fossil fuel extracted may increase total $E_{FF}$ uncertainty up to 11%. Even considering an uncertainty range as high as the one suggested by Quilcaille et al. (2016), the difference would be 0.15PgC/yr, which would not suffice to explain the $CO_2$ stabilization in the 1940s. The authors, nevertheless, agree that it is worth including a note about the subject in the revised version of the manuscript.

We now include a reference to this problem in Section 2.2.1 (page 5, lines 28-30):

> *"However, it should be noted that an uncertainty range of about 11% may be more realistic when accounting for differences in the datasets and methods to estimate CO2 emissions (Mohr et al., 2015; Quilcaille et al., 2016)."*

**RC: However, I have reported several specific comments to improve the paper in the attachment.**
AR: We have included the corrections proposed and changed the figures and tables accordingly.

Mohr, S. H., et al. "Projection of world fossil fuels by country." *Fuel* 141 (2015): 120-135.

Quilcaille, Yann, et al. "Uncertainty in projected climate change caused by methodological discrepancy in estimating CO2 emissions from fossil fuel combustion." *EGU General Assembly Conference Abstracts*. Vol. 18. 2016.

**The anomalous 1940's dip in CO2 was noted twenty years ago and a definitive explanation is still lacking. By presenting an extensive range of comparisons, this paper brings into focus the difficulties. As such, it is a valuable contribution and, subject to clarifying the issues noted below, is suitable for publication in Biogeosciences.**
The authors would like to thank the referee for the review and for highlighting important aspects that need improvement.

**RC1: I would have liked a clearer statement, for each of the comparisons, of the "boundary conditions" applicable for each case, i.e. what is being assumed as "fixed" in each case (e.g. single deconvolutions assume fixed (i.e. time-invariant) ocean response, while double deconvolutions assume invariant mixed-layer response).**
AR: We agree with the referee and this point will is now included in page 3 (lines 17-18):

> "Single deconvolutions do not use the $\delta^{13}C$ information and assume time-invariant ocean response."

And in page 7 (lines 2-5):

> "Their analysis relied on a previous dataset (Etheridge et al., 1996; Francey et al., 1999) of the same CO2 ice-core record used here (Rubino et al., 2013) to solve two mass-balance equations for atmospheric $CO_2$ and $\delta^{13}C$, assuming fixed ocean mixed-layer response."

**RC2: With regard to the results presented in fig 2, the authors note that they are comparing (by taking the difference) AGR based on a 25-yr cutoff spline and AGR based on splines with a longer cutoff. This can be simply described as applying a band-pass filter to AGR. Saying this explicitly might help the reader, but it also suggests that the analysis in figure 2 adds little to the overall analysis in the paper.**
AR: The purpose of Fig. 2 is two-fold:
   i)     To present the estimates of the ocean sink from the double-deconvolution by Joos et al. (1999), which are then used as a reference for the possible contribution of the ocean to the increased CO2 uptake during the 1940s;
   ii)    To show that discrepancies between AGR reconstructed from the different terms and the observations are still to be expected, partly because of the different choices of smoothing, but also for the reasons discussed in Section 2.3.1.
Therefore, we consider that Fig. 2 is worth keeping, and have added the following sentence in page 7 (lines 26-27) in order to clarify the use of Joos et al. (1999) dataset:

> "Here we use estimates of the ocean sink from the double-deconvolution by Joos et al. (1999), ie. $O_J$ , as a reference for natural variability in the ocean sink."

**RC3: A minor point is the implication that zero AGR requires zero fossil emissions (or a change on uptake processes). This is not correct. Zero growth rate can be achieved by a rapid reduction in emissions, with uptake processes responding to higher atmospheric concentrations. Note for example stabilisation calculations, or the discussion by Gloor 2010 (Atmos Phys Chem, 10, 7739). (This a case of poor wording and it in no way invalidates the overall analysis in the paper)**
AR: We thank the referee for pointing this important issue, as the phrasing was indeed not the most correct. However, given the datasets available it is not likely that an abrupt decrease in $E_{FF}$ as large as the one

needed to stabilize atmospheric $CO_2$ might have occurred during the 1940s (see AR to Referee #2). Nevertheless, we agree that the sentence can be reformulated, and now reads (page 2, lines 26-28):

[revised manuscript text omitted]

**Figure 7.** Resulting E$_{LUC}$ from OSCAR simulations for hypothetical scenarios about changes in LUC during 1940-1950 (top). In T1 forest conversion is set to zero (green solid), in T2 the rate of forest expansion during the period is doubled (green dashed), in T3 cropland and pasture expansion are stopped (yellow solid) and in T4 wood harvest is set to zero (yellow dashed). The E$_{LUC}$ from each test are compared with the LUC emissions in the standard OSCAR simulation (red). The atmospheric $CO_2$ growth-rate (AGR) resulting from standard OSCAR and each test are compared with the ice-core record (center). The $\delta^{13}C$ values corresponding to each test (bottom) are compared with $\delta^{13}C$ from the ice-core record and the corresponding uncertainty (markers and errorbars).